# Sharpness-Aware Training for Free

**Jiawei Du**[1,2] , **Daquan Zhou**[3] , **Jiashi Feng**[3] , **Vincent Y. F. Tan**[4,2] , **Joey Tianyi Zhou**[1,2]

[1]Centre for Frontier AI Research (CFAR), A*STAR, Singapore,
[2]Department of Electrical and Computer Engineering, National University of Singapore
[3]ByteDance
[4]Department of Mathematics, National University of Singapore

## Abstract

Modern deep neural networks (DNNs) have achieved state-of-the-art performances but are typically over-parameterized. The over-parameterization may result in undesirably large generalization error in the absence of other customized training strategies. Recently, a line of research under the name of *Sharpness-Aware Minimization* (SAM) has shown that minimizing a sharpness measure, which reflects the geometry of the loss landscape, can significantly reduce the generalization error. However, SAM-like methods incur a two-fold computational overhead of the given base optimizer (e.g. SGD) for approximating the sharpness measure. In this paper, we propose *Sharpness-Aware Training for Free*, or *SAF*, which mitigates the sharp landscape at *almost zero additional computational cost* over the base optimizer. Intuitively, SAF achieves this by avoiding sudden drops in the loss in the sharp local minima throughout the trajectory of the updates of the weights. Specifically, we suggest a novel trajectory loss, based on the KL-divergence between the outputs of DNNs with the current weights and past weights, as a replacement of the SAM's sharpness measure. This loss captures the rate of change of the training loss along the model's update trajectory. By minimizing it, SAF ensures the convergence to a flat minimum with improved generalization capabilities. Extensive empirical results show that SAF minimizes the sharpness in the same way that SAM does, yielding better results on the ImageNet dataset with essentially the same computational cost as the base optimizer. Our codes are available at https://github.com/AngusDujw/SAF.

## 1 Introduction

Despite achieving remarkable performances in many applications, powerful neural networks [2, 5, 21, 26, 26, 27, 34, 35] are typically over-parameterized. Such over-parameterized deep neural networks require advanced training strategies to ensure that their generalization errors are appropriately small [2, 32] and the adverse effects of the overfitting are alleviated. Understanding how deep neural networks generalize is perhaps the most fundamental and yet perplexing topic in deep learning.

Numerous studies expend significant amounts of efforts to understand the generalization capabilities of deep neural networks and mitigate this problem from a variety of perspectives, such as the information perspective [18], the model compression perspective [1, 9], and the Bayesian perspective [22, 24], etc. The loss surface geometry perspective, in particular, has attracted a lot of attention from researchers recently [4, 12–14, 28]. These studies connect the generalization gap and the sharpness of the minimum's loss landscape, which can be characterized by the largest eigenvalue of the Hessian matrix $\nabla_\theta^2 f_\theta$ [14] where $f_\theta$ represents the input-output map of the neural network. In other words, a (local) minimum that is located in a flatter region tends to generalize better than one that is located in a sharper one [4, 13]. The recent work [8] proposes an effective and generic training algorithm, named *Sharpness-Aware Minimization* (SAM), to encourage the training process to converge to a flat

36th Conference on Neural Information Processing Systems (NeurIPS 2022).

minimum. SAM explicitly penalizes a sharpness measure to obtain flat minima, which has achieved state-of-the-art results in several learning tasks [2, 36].

Unfortunately, SAM's computational cost is twice that compared to the given base optimizer, which is typically stochastic gradient descent (SGD). This prohibits SAM from being deployed extensively in highly over-parameterized networks. Half of SAM's computational overhead is used to approximate the sharpness measure in its first update step. The other half is used by SAM to minimize the sharpness measure together with the vanilla loss in the second update step. As shown in Figure 1, SAM significantly reduces the generalization error at the expense of double computational overhead.

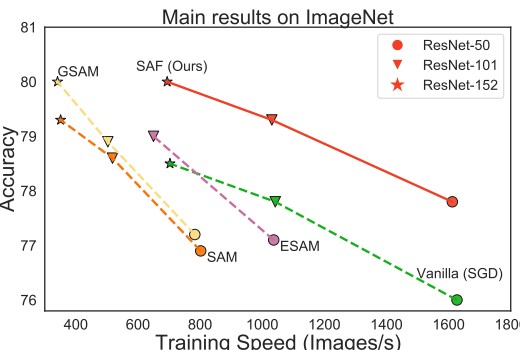

Figure 1: Accuracy vs training speed of SGD, SAM [8], ESAM [6], GSAM [36], and SAF. GSAM is the state-of-the-art among SAM's follow-up works. Every connected line represents a method that trains ResNet-50, ResNet-101, and ResNet-152 models on ImageNet-1k. SAF outperforms SAM and its variants yet requires *no additional computational overhead.*

Liu *et al.* [20] and Du *et al.* [6] recently addressed the computation issue of SAM and proposed LookSAM [20] and Efficient SAM (ESAM) [6], respectively. LookSAM only minimizes the sharpness measure *once* in the first of every five iterations. For the other four iterations, LookSAM reuses the gradients that minimizes the sharpness measure, which is obtained by decomposing the SAM's updated gradients in the first iteration into two orthogonal directions. As a result, LookSAM saves $40\%$ computations compared to SAM but unfortunately suffers from performance degradation. On the other hand, ESAM proposes two strategies to save the computational overhead in SAM's two updating steps. The first strategy approximates the sharpness by using fewer weights for computing; the other approximates the updated gradients by only using a subset containing the instances that contribute the most to the sharpness. ESAM is reported to save up to $30\%$ computations without degrading the performance. However, ESAM's efficiency degrades in large-scale datasets and architectures (from $30\%$ on CIFAR-10/100 to $22\%$ on ImageNet). LookSAM and ESAM both follow SAM's path to minimize SAM's sharpness measure, which limits the potential for further improvement of their efficiencies.

In this paper, we aim to perform sharpness-aware training with *zero additional computations and yet still retain superior generalization performance*. Specifically, we introduce a novel trajectory loss to replace SAM's sharpness measure loss. This trajectory loss measures the KL-divergence between the outputs of neural networks with the current weights and those with the past weights. We propose the *Sharpness-Aware training for Free* (SAF) algorithm to penalize the trajectory loss for sharpness-aware training. More importantly, SAF requires *almost zero extra computations* (SAF 0.7% v.s. SAM 100%). SAF memorizes the outputs of DNNs in the process of training as the targets of the trajectory loss. By minimizing it, SAF avoids the quick converging to a local sharp minimum. SAF has the potential to result in out-of-memory issue on extremely large scale datasets, such as ImageNet-21K [15] and JFT-300M [25]. We also introduce a memory-efficient variant of SAF, which is *Memory-Efficient Sharpness-Aware Training* (MESA). MESA adopts a DNN whose weights are the exponential moving averages (EMA) of the trained DNN to output the targets of the trajectory loss. As a result, MESA resolves the out-of-memory issue on extremely large scale datasets, at the cost of 15% additional computations (v.s. SAM 100%). As shown in Figure 2, SAF and MESA both encourage the training to converge to flat minima similarly as SAM. Besides visualizing the loss landscape, we conduct experiments on the CIFAR-10/100 [16] and the ImageNet [3] datasets to verify the effectiveness of SAF. The experimental results indicate that our proposed SAF and MESA outperform SAM and its variants with almost twice the training speed; this is illustrated on the ImageNet-1k dataset in Figure 1.

In a nutshell, we summarize our contributions as follows.

- We propose a novel trajectory loss to measure the sharpness to be used for sharpness-aware training. Requiring almost zero extra computational overhead, this trajectory loss is a better loss to quantify the sharpness compared to SAM's loss.

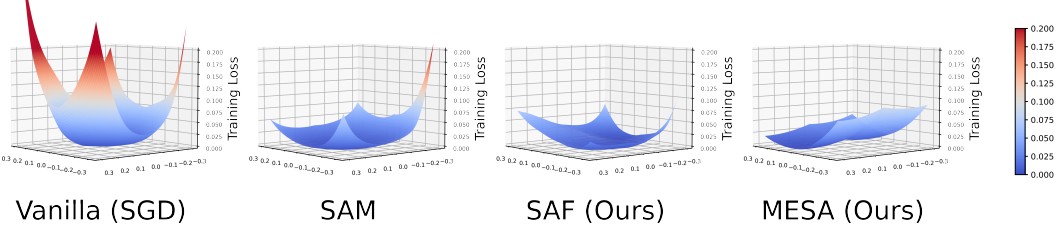

| Vanilla (SGD) | SAM | SAF (Ours) | MESA (Ours) |

Figure 2: Visualizations of loss landscapes [2, 17] of the PyramidNet-110 model on the CIFAR-100 dataset trained with SGD, SAM, our proposed SAF, and MESA. SAF encourages the networks to converge to a flat minimum with zero additional computational overhead.

- We address the efficiency issue of current sharpness-aware training, which generally incurs twice the computational overhead compared to regular training. Based on our proposed trajectory loss, we propose the novel SAF algorithm for improved generalization ability in this paper. SAF is demonstrated to outperform SAM on the ImageNet dataset, with the same computational cost as the base optimizer.

- We also propose the MESA algorithm as a memory-efficient variant of SAF. MESA reduces the extra memory-usage of SAF at the cost of 15% additional computations, which allows SAF/MESA to be deployed efficiently (both in terms of memory and computation) on extremely large-scale datasets (e.g. ImageNet-21K [15]).

## 2 Preliminaries

Throughout the paper, we use $f_\theta$ to denote a neural network with weight parameters $\theta$. We are given a training dataset $\mathbb{S}$ that contains i.i.d. samples drawn from a natural distribution $\mathcal{D}$. The training of the network is typically a non-convex optimization problem which aims to search for an optimal weight vector $\hat{\theta}$ that satisfies

$$\hat{\theta} = \arg\min_\theta L_\mathbb{S}(f_\theta) \quad \text{where} \quad L_\mathbb{S}(f_\theta) = \frac{1}{|\mathbb{S}|} \sum_{x_i \in \mathbb{S}} \ell(f_\theta(x_i)), \tag{1}$$

where $\ell$ can be an arbitrary loss function, and we use $x_i$ to denote the pair (inputs, targets) of the $i$-th element in the training set. In this paper, we take $\ell$ to be the cross entropy loss; We use $\|\cdot\|$ represents $\ell_2$ norm; *we assume that $L_\mathbb{S}(f_\theta)$ is continuous and differentiable, and its first-order derivation is bounded*. In each training iteration, optimizers randomly sample a mini-batch $\mathbb{B}_t \subset \mathbb{S}$ with a fixed batch size.

**Sharpness-Aware Minimization** The conventional optimization and training focuses on minimizing the empirical loss of a single weight vector $\hat{\theta}$ over the training set $\mathbb{S}$ as stated in Equation 1. This training paradigm is known as *empirical risk minimization*, and tends to overfit to the training set and converges to sharp minima. Sharpness-Aware Minimization (SAM) [8] aims to encourage the training to converge to a flatter region in which the training losses in the neighborhood around the minimizer $\hat{\theta}$ are lower. To achieve this, SAM proposes a training scheme that solves the following min-max optimization problem:

$$\min_\theta \max_{\epsilon:\|\epsilon\|_2 \leq \rho} L_\mathbb{S}(f_{\theta+\epsilon}). \tag{2}$$

where $\rho$ is a predefined constant that constrains the radius of the neighborhood; $\epsilon$ is the weight perturbation vector that maximizes the training loss within the $\rho$-constrained neighborhood. The objective loss function of SAM can be rewritten as the sum of the vanilla loss and the loss associated to the sharpness measure, which is the maximized change of the training loss within the $\rho$-constrained neighborhood, i.e.,

$$\hat{\theta} = \arg\min_\theta \left\{ R_\mathbb{S}(f_\theta) + L_\mathbb{S}(f_\theta) \right\} \quad \text{where} \quad R_\mathbb{S}(f_\theta) = \max_{\epsilon:\|\epsilon\|_2 \leq \rho} L_\mathbb{S}(f_{\theta+\epsilon}) - L_\mathbb{S}(f_\theta). \tag{3}$$

The sharpness measure is approximated as $R_\mathbb{S}(f_\theta) = L_\mathbb{S}(f_{\theta+\hat{\epsilon}}) - L_\mathbb{S}(f_\theta)$, where $\hat{\epsilon}$ is the solution to an approximated version of the maximization problem where the objective is the first-order Taylor series approximation of $L_\mathbb{S}(f_{\theta+\epsilon})$ around $\theta$, i.e.,

$$\hat{\epsilon} = \arg\max_{\epsilon:\|\epsilon\|_2 < \rho} L_\mathbb{S}(f_{\theta+\epsilon}) \approx \rho \frac{\nabla_\theta L_\mathbb{S}(f_\theta)}{\|\nabla_\theta L_\mathbb{S}(f_\theta)\|}. \tag{4}$$

Intuitively, SAM seeks flat minima with low variation in their training losses when the optimal weights are slightly perturbed.

## 3 Methodology

The fact that SAM's computational cost is twice that compared to the base optimizer is its main limitation. The additional computational overhead is used to compute the sharpness term $R_{\mathbb{S}}(f_\theta)$ in Equation 3. We propose a new trajectory loss as a replacement of SAM's sharpness loss $R_{\mathbb{S}}(f_\theta)$ with essentially zero extra computational overhead over the base optimizer. Next, we introduce the *Sharpness-Aware Training for Free* (SAF) algorithm whose pseudocode can be found in Algorithm 1. We first start with recalling SAM's sharpness measure loss. Then we explain the intuition for the trajectory loss as a substitute for SAM's sharpness measure loss in Section 3.1. Next, we present the complete algorithm of SAF in Section 3.2 and a memory-efficient variant of it in Section 3.3.

---

**Algorithm 1** Training with SAF and MESA

---

**Input:** Training set $\mathbb{S}$; A network $f_\theta$ with weights $\theta$; Learning rate $\eta$; Epochs $E$; Iterations $T$ per epoch; SAF starting epoch $E_{\text{start}}$; SAF coefficients $\lambda$; Temperature $\tau$; SAF hyperparameter $\tilde{E}$; EMA decay factor $\beta$ for MESA.

1: **for** $e = 1$ to $E$ **do**           ▷ $e$ represents the current epoch
2:   **for** $t = 1$ to $T$, Sample a mini-batch $\mathbb{B} \subset \mathbb{S}$ **do**
3:    **if** SAF **then**
4:     Record the outputs: $\hat{y}_i^e \leftarrow f_\theta(x_i)$, where $x_i \in \mathbb{B}$
5:     Load $\hat{y}_i^{(e-\tilde{E})}$ saved in $\tilde{E}$ epochs ago for each $x_i \in \mathbb{B}$
6:     Compute $L_{\mathbb{B}}^{\text{tra}}(f_\theta, \mathbb{Y}^{(e-\tilde{E})})$      ▷ Defined in Equation 9;
7:    **else if** MESA **then**
8:     Update EMA model weights: $v_t = \beta v_{t-1} + (1-\beta)\theta$
9:     Compute $L_{\mathbb{B}}^{\text{tra}}(f_\theta, f_{v_t})$        ▷ Defined in Equation 13
10:    **if** $e > E_{\text{start}}$ **then**     ▷ Added the trajectory loss after $E_{\text{start}}$ epoch
11:     $\mathcal{L} = L_{\mathbb{B}}(f_\theta) + \lambda L_{\mathbb{B}}^{\text{tra}}$
12:    **else**
13:     $\mathcal{L} = L_{\mathbb{B}}(f_\theta)$
14:    Update the weights: $\theta \leftarrow \theta - \eta \nabla_\theta \mathcal{L}$

**Output:** A flat minimum solution $\tilde{\theta}$.

---

### 3.1 General Idea: Leverage the trajectory of weights to estimate the sharpness

We first rewrite the sharpness measure $R_{\mathbb{B}}(f_\theta)$ of SAM based on its first-order Taylor expansion. Given a vector $\hat{\epsilon}$ (Equation 4) whose norm is small, we have

$$R_{\mathbb{B}}(f_\theta) = L_{\mathbb{B}}(f_{\theta+\hat{\epsilon}}) - L_{\mathbb{B}}(f_\theta) \approx L_{\mathbb{B}}(f_\theta) + \hat{\epsilon}\nabla_\theta L_{\mathbb{B}}(f_\theta) - L_{\mathbb{B}}(f_\theta)$$
$$= \rho \frac{\nabla_\theta L_{\mathbb{B}}(f_\theta)^\top}{\|\nabla_\theta L_{\mathbb{B}}(f_\theta)\|} \nabla_\theta L_{\mathbb{B}}(f_\theta) = \rho \|\nabla_\theta L_{\mathbb{B}}(f_\theta)\|. \tag{5}$$

We remark that minimizing the sharpness loss $R_{\mathbb{B}}(f_\theta)$ is equivalent to minimizing the $\ell_2$-norm of the gradient $\nabla_\theta L_{\mathbb{B}}(f_\theta)$, which is the same gradient used to minimize the vanilla loss $L_{\mathbb{B}}(f_\theta)$.

The learning rate $\eta$ in the standard training (using SGD) is typically smaller than $\rho$ as suggested by SAM [8]. Hence, if the mini-batch $\mathbb{B}$ is the same for the two consecutive iterations, the change of the training loss after the weights have been updated can be approximated as follow,

$$L_{\mathbb{B}}(f_\theta) - L_{\mathbb{B}}(f_{\theta-\eta\nabla_\theta L_{\mathbb{B}}(f_\theta)}) \approx \eta \|\nabla_\theta L_{\mathbb{B}}(f_\theta)\|^2 \approx \frac{\eta}{\rho^2} R_{\mathbb{B}}(f_\theta)^2. \tag{6}$$

We also remark that the change of the training loss after the weights have been updated is proportional to $R_{\mathbb{B}}(f_\theta)^2$. Hence, minimizing the loss difference is equal to minimizing the sharpness loss of SAM in this case.

This inspires us to leverage the update of the weights in the standard training to approximate SAM's sharpness measure. Regrettably though, the samples in the mini-batches $\mathbb{B}_t$ and $\mathbb{B}_{t+1}$ in two consecutive iterations are different with high probability, which does not allow the sharpness to be computed as in Equation 6. This is precisely the reason why SAM uses an additional step to compute $\hat{\epsilon}$ for approximating the sharpness. To avoid these additional computations completely, we introduce a novel trajectory loss that makes use of the trajectory of the weights learned in the standard training procedure to measure sharpness.

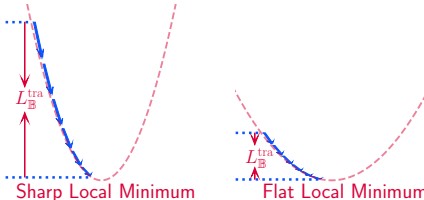

Figure 3: For the vanilla training loss $L_{\mathbb{B}}(f_\theta)$ (dashed lines), the blue arrows represent the trajectory during training. **Left**: A sharp local minimum tends to have a large trajectory loss. **Right**: By minimizing the trajectory loss, SAF prevents the training from converging to a sharp local minimum.

We denote the past trajectory of the weights as the iterations progress as $\Theta = \{\theta_2, \theta_3, \dots, \theta_{t-1}\}$. Hence, $\theta_t$ represents the weights in the $t$-th iteration, which is also the current model update step. We use SGD as the base optimizer to illustrate our ideas. Recall that standard SGD updates the weights as $\theta_{t+1} = \theta_t - \eta \nabla_{\theta_t} L_{\mathbb{B}_t}(f_{\theta_t})$. We aim to derive an equivalent term of the sharpness $R_{\mathbb{B}_t}(f_{\theta_t})$ that can be computed **without additional computations**. The quantity $R_{\mathbb{B}_t}(f_{\theta_t})$ as defined in Equation 5 is always non-negative, and thus we have

$$
\begin{aligned}
\underset{\theta_t}{\arg\min} R_{\mathbb{B}_t}(f_{\theta_t}) &= \underset{\theta_t}{\arg\min} \gamma_t R_{\mathbb{B}_t}(f_{\theta_t}) R_{\mathbb{B}_t}(f_{\theta_t}) \\
&= \underset{\theta_t}{\arg\min}[\gamma_t R_{\mathbb{B}_t}(f_{\theta_t}) R_{\mathbb{B}_t}(f_{\theta_t}) + \gamma_i R_{\mathbb{B}_t}(f_{\theta_i}) R_{\mathbb{B}_i}(f_{\theta_i})] \\
&= \underset{\theta_i \sim \mathrm{Unif}(\Theta)}{\mathbb{E}}[\gamma_i R_{\mathbb{B}_t}(f_{\theta_i}) R_{\mathbb{B}_i}(f_{\theta_i})],
\end{aligned}
\tag{7}
$$

where $\theta_i \sim \mathrm{Unif}(\Theta)$ means that $\theta_i$ is uniformly distributed in the set $\Theta$, $\gamma_i$ is defined as $\frac{\eta_t}{\rho^2}\cos(\Phi_i)$, where $\Phi_i$ is the angle between the gradients that are computed using the mini-batches $\mathbb{B}_i$ and $\mathbb{B}_t$, respectively. Note that for $i \neq t$, $\gamma_i R_{\mathbb{B}_t}(f_{\theta_i}) R_{\mathbb{B}_i}(f_{\theta_i})$ is a constant with respect to the variable $\theta_t$. Therefore, when we traverse all the terms that satisfy $i \neq t$, we obtain Equation 7. Hence, the sharpness can therefore be alternatively estimated as follows:

$$
\begin{aligned}
\underset{\theta_i \sim \mathrm{Unif}(\Theta)}{\mathbb{E}}[\gamma_i R_{\mathbb{B}_t}(f_{\theta_i}) R_{\mathbb{B}_i}(f_{\theta_i})] &\approx \underset{\theta_i \sim \mathrm{Unif}(\Theta)}{\mathbb{E}}\left[\eta_i \cos(\Phi_i)\|\nabla_{\theta_i} L_{\mathbb{B}_t}(f_{\theta_i})\| \|\nabla_{\theta_i} L_{\mathbb{B}_i}(f_{\theta_i})\|\right] \\
&= \underset{\theta_i \sim \mathrm{Unif}(\Theta)}{\mathbb{E}}\left[\eta_i \nabla_{\theta_i} L_{\mathbb{B}_t}(f_{\theta_i})^\top \nabla_{\theta_i} L_{\mathbb{B}_i}(f_{\theta_i})\right] \approx \underset{\theta_i \sim \mathrm{Unif}(\Theta)}{\mathbb{E}}\left[L_{\mathbb{B}_t}(f_{\theta_i}) - L_{\mathbb{B}_t}(f_{\theta_{i+1}})\right] \\
&= \frac{1}{t-1}\left[L_{\mathbb{B}_t}(f_{\theta_1}) - L_{\mathbb{B}_t}(f_{\theta_t})\right].
\end{aligned}
\tag{8}
$$

Further details of the above derivation can be found in Appendix A.1. We remark that minimizing the loss difference $L_{\mathbb{B}_t}(f_{\theta_1}) - L_{\mathbb{B}_t}(f_{\theta_t})$ is equivalent to minimizing the SAM's sharpness measure $R_{\mathbb{B}_t}(f_{\theta_t})$. Accordingly, requiring no additional computational overhead, the training loss difference is a good proxy to SAM's loss $R_{\mathbb{B}_t}(f_{\theta_t})$ to quantify and penalize the sharpness.

## 3.2 Sharpness-Aware Training for Free (SAF)

We elaborate our proposed training algorithm SAF in this subsection. To estimate the sharpness precisely, SAF only takes the update trajectory in the past $\tilde{E}$ epochs into consideration. When simultaneously minimizing the vanilla loss and the training loss difference (as in Equation 8), the second term $-L_{\mathbb{B}_T}(f_{\theta_T})$ will unfortunately cancel out the vanilla loss. Therefore, we replace the cross entropy loss with the Kullback–Leibler (KL) divergence loss to decouple the vanilla loss. We also soften the targets of the KL divergence loss using a temperature $\tau$. Accordingly, the trajectory loss at $e$-th epoch is defined as follow

$$
L_{\mathbb{B}}^{\mathrm{tra}}(f_\theta, \mathbb{Y}^{(e-\tilde{E})}) = \frac{\lambda}{|\mathbb{B}|} \sum_{x_i \in \mathbb{B}, \hat{y}_i^{(e-\tilde{E})} \in \mathbb{Y}^{(e-\tilde{E})}} \mathrm{KL}\left(\frac{1}{\tau}\hat{y}_i^{(e-\tilde{E})}, \frac{1}{\tau}f_\theta(x_i)\right),
\tag{9}
$$

where $\mathbb{Y}^{(e-\tilde{E})} = \{\hat{y}_i^{(e-\tilde{E})} = f_\theta^{(e-\tilde{E})}(x_i) : x_i \in \mathbb{B}\}$. We remark that $\hat{y}_i^{(e-\tilde{E})}$ is the network output of the instance $x_i$ in $\tilde{E}$ epochs ago, as illustrated in Line 4 of Algorithm 1. The outputs of each

instance $x_i$ will be recorded, and no additional computations are required during this procedure. The trajectory loss will be deployed after a predefined epoch $E_{\text{start}}$ (Line 10 of Algorithm 1), because the outputs of the DNN are neither stable nor reliable at the first few epochs. Intuitively, the trajectory loss slows down the rate of change of the training loss to avoid convergence to sharp local minima. This is illustrated in Figure 3.

### 3.3 Memory-Efficient Sharpness-Aware Training (MESA)

The memory usage for recording the outputs is negligible for standard datasets such as CIFAR [16] (57 MB). However, SAF's memory usage is proportional to the size of the training datasets, which may result in an out-of-memory issue on extremely large-scale datasets such as ImageNet-21k [15] and JFT-300M [25]. Another limitation of SAF is that the coefficient $\gamma_t$ will decay at the same rate as the learning rate $\eta_t$ since $\gamma_t \propto \eta_t$ as shown in Equation 8. Hence, the sharpness at the current weights are quantified by smaller coefficients $\gamma_t$. For a SAF-like algorithm to be applicable on extremely large-scale datasets and to emphasize the sharpness of the *most current or recent* weights, we introduce *Memory-Efficient Sharpness-Aware Training* (MESA), which adopts an exponential moving average (EMA) weighting strategy on the weights to construct the trajectory loss. The *EMA weight* $v_t$ at the $t$-th iteration is updated as follows

$$v_t = \beta v_{t-1} + (1-\beta)\theta_t, \tag{10}$$

where $\beta \in (0.9, 1)$ is the decay coefficient of EMA. Given that $v_1 = \theta_1$, and $\theta_{t+1} = \theta_t - \eta\nabla_{\theta_t}L_{\mathbb{B}_t}(f_{\theta_t})$, the EMA weight in the $t$-th iteration, can be expressed as

$$v_t = \theta_1 - \sum_{i=1}^{t-1}(1-\beta^{t-i})\eta\nabla_{\theta_i}L_{\mathbb{B}_i}(f_{\theta_i}) = \theta_t + \sum_{i=1}^{t-1}\beta^{t-i}\eta\nabla_{\theta_i}L_{\mathbb{B}_i}(f_{\theta_i}). \tag{11}$$

More details of this derivation can be found in Appendix A.2. Therefore, the trajectory from $\theta_t$ to $v_t$ is collected in the vector $\mathbb{W}_{\text{EMA}} = (w_2, w_3, \ldots, w_{t-1})$, where $w_i = w_{i-1} - \beta^{t-i}\eta\nabla_{\theta_i}L_{\mathbb{B}_i}(f_{\theta_i})$, $w_1 = v_t$, and $w_t = \theta_t$. If we regard the outputs of EMA model $f_{v_t}$ as the targets of the trajectory loss, and substitute it into Equation 8,

$$\frac{1}{t-1}\left[L_{\mathbb{B}_t}(f_{v_t}) - L_{\mathbb{B}_t}(f_{\theta_t})\right] = \mathop{\mathbb{E}}_{w_i \sim \text{Unif}(\mathbb{W}_{\text{EMA}})}\left[L_{\mathbb{B}_t}(f_{w_i}) - L_{\mathbb{B}_t}(f_{w_{i+1}})\right]$$
$$\approx \mathop{\mathbb{E}}_{w_i \sim \text{Unif}(\mathbb{W}_{\text{EMA}})}\left[\beta^{t-i}\gamma_i R_{\mathbb{B}_t}(f_{w_i})R_{\mathbb{B}_t}(f_{\theta_i})\right]. \tag{12}$$

More details can be found in Appendix A.1. Hence, the EMA coefficients $\beta^{t-i}$ will place more emphasis on the sharpness of the *current* and *most recent* weights since $0 < \beta < 1$. The trajectory loss of MESA is

$$L_{\mathbb{B}}^{\text{tra}}(f_\theta, f_{v_t}) = \frac{1}{|\mathbb{B}|}\sum_{x_i \in \mathbb{B}}\text{KL}\left(\frac{1}{\tau}f_{v_t}(x_i), \frac{1}{\tau}f_\theta(x_i)\right). \tag{13}$$

We see that the difference between this and the trajectory loss $L_{\mathbb{B}}^{\text{tra}}(f_\theta, \mathbb{Y}^{(e-\tilde{E})})$ discussed in Section 3.2 is that the target $\hat{y}_i^{(e-\tilde{E})}$ at the $t$-iteration has been replaced by $f_{v_t}(x_i)$ for $x_i \in \mathbb{B}$.

## 4 Experiments

We verify the effectiveness of our proposed SAF and MESA algorithms in this section. We first conduct experiments to demonstrate that our proposed SAF achieves better performance compared to SAM which requires twice the training speed. SAF is shown to outperform SAM and its variants in large-scale datasets and models. The main results are summarized into Tables 1 and 3. Next, we evaluate the sharpness of SAF using the measurement proposed by SAM. We demonstrate that SAF encourages the training to converge to a flat minimum. We also visualize the loss landscape of the minima converged by SGD, SAM, SAF, and MESA in Figures 2 and 5, which show that both SAF's and MESA's loss landscapes are as flat as SAM.

## 4.1 Experiment Setup

**Datasets** We conduct experiments on the following image classification benchmark datasets: CIFAR-10, CIFAR-100 [16], and ImageNet [3]. The 1000-class ImageNet dataset contains roughly 1.28 million training images, which is the popular benchmark for evaluating large-scale training.

**Models** We employ a variety of widely-used DNN architectures to evaluate the performance and training speed. We use ResNet-18 [11], Wide ResNet-28-10 [31], and PyramidNet-110 [10] for the training in CIFAR-10/100 datasets. We use ResNet [11] and Vision Transformer [5] models with various sizes on the ImageNet dataset.

**Baselines** We take the vanilla (AdamW for ViT, SGD for the other models), SAM [8], ESAM [6], GSAM [36], and LookSAM [20] as the baselines. ESAM and LookSAM are SAM's follow-up works that improve efficiency. GSAM achieves the best performance among the SAM's variants. We reproduce the results of the vanilla and SAM, which matches the reported results in [2, 20, 36]. And we report the cited results of baselines ESAM [6], GSAM [36], and LookSAM [20].

**Implementation details** We set all the training hyperparameters to be the same for a fair comparison among the baselines and our proposed algorithms. The details of the training setting are displayed in the Appendix. We follow the settings of [2, 6, 20, 36] for the ImageNet datasets, which is different from the experimental setting of the original SAM paper [8]. The codes are implemented based on the TIMM framework [29]. The ResNets are trained with a batch size of 4096, 1.4 learning rate, 90 training epochs, and SGD optimizer (momentum=0.9) over 8 Nvidia V-100 GPU cards. The ViTs are trained with 300 training epochs and AdamW optimizer ($\beta_1 = 0.9, \beta_2 = 0.999$). We only conduct the basic data augmentation for the training on both CIFAR and ImageNet (Inception-style data augmentation). The hyperparameters of SAF and MESA are consistent among various DNNs architectures and various datasets. We set $\tau = 5, \tilde{E} = 3, E_{\text{start}} = 5$ for all the experiments, $\lambda \in \{0.3, 0.8\}$ for SAF and MESA , respectively.

Table 1: Classification accuracies and training speeds on the ImageNet dataset. The numbers in parentheses $(\cdot)$ indicate the ratio of the training speed w.r.t. the vanilla base optimizer's (SGD's) speed. Green indicates improvement compared to SAM, whereas red suggests a degradation.

| ImageNet | ResNet-50 | | ResNet-101 | |
| --- | --- | --- | --- | --- |
| | Accuracy | images/s | Accuracy | images/s |
| Vanilla (SGD) | 76.0 | 1,627 (100%) | 77.8 | 1,042 (100%) |
| SAM [8] | 76.9 | 802 (49.3%) | 78.6 | 518 (49.7%) |
| ESAM [1] [6] | 77.1 | 1,037 (63.7%) | 79.1 | 650 (62.4%) |
| GSAM [2] [36] | 77.2 | 783 (48.1%) | 78.9 | 503 (48.3%) |
| SAF (Ours) | **77.8** | 1,612 (99.1%) | **79.3** | 1,031 (99.0%) |
| MESA (Ours) | 77.5 | 1,386 (85.2%) | 79.1 | 888 (85.4%) |

| ImageNet | ResNet-152 | | ViT-S/32 | |
| --- | --- | --- | --- | --- |
| | Accuracy | images/s | Accuracy | images/s |
| Vanilla [3] | 78.5 | 703 (100%) | 68.1 | 5,154 (100%) |
| SAM [8] | 79.3 | 351 (49.9%) | 68.9 | 2,566 (49.8%) |
| LookSAM [4] [20] | - | - | 68.8 | 4,273 (82.9%) |
| GSAM [2] [36] | **80.0** | 341 (48.5%) | **73.8** | 2,469 (47.9%) |
| SAF (Ours) | 79.9 | 694 (98.7%) | 69.5 | 5,108 (99.1%) |
| MESA (Ours) | **80.0** | 601 (85.5%) | 69.6 | 4,391 (85.2%) |

## 4.2 Experimental Results

**ImageNet** Our proposed SAF and MESA achieve better performance on the ImageNet dataset compared to the other competitors. We report the best test accuracies and the average training speeds

---

[1]We report in [6], as ESAM only release their codes in Single-GPU environment.

[2]We report the results in [36], but failed to reproduce them using the officially released codes.

[3]We use base optimizers SGD for ResNet-152 and AdamW for ViT-S/32.

[4]The authors of LookSAM have not released their code for either Single- or Multi-GPU environments; hence we report the results in [20]. LookSAM only reports results for ViTs and there are no results for ResNet.

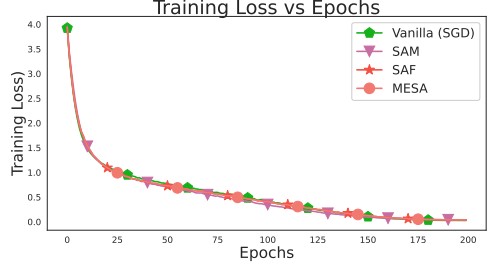
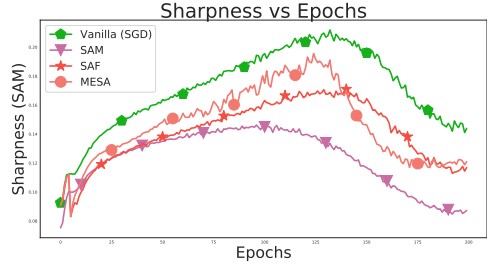

(a) Training loss vs Epochs of SAF.          (b) The SAM's sharpness measure vs epochs

Figure 4: **Left**: The change of the vanilla loss (exclude the trajectory loss) in each epoch. SAF does not affect the convergence of the training. **Right**: The change of sharpness, which is the measurement proposed by SAM with $\rho = 0.05$. SAF and MESA decrease the sharpness measure of SAM significantly.

in Table 1. The experiment results demonstrate that SAF incurs no additional computational overhead to ensure that the training speed is the same as the base optimizer—SGD. We observe that SAF and MESA perform better in the large-scale datasets. SAF trains DNN at a douled speed than SAM (SAF 99.2% vs SAM 50%). MESA achieves a training speed which is 84% to 85% that of SGD. Concerning the performance, both SAF and MESA outperform SAM up to (0.9%) on the ImageNet dataset. More importantly, SAF and MESA achieve a new state-of-the-art results of SAM's follow-up works on the ImageNet datasets trained with ResNets.

**CIFAR10 and CIFAR100** We ran all the experiments three times with different random seeds for fair comparisons. We summarize the results in Table 3, in the same way as we do for the experiments on the ImageNet dataset. The training speed is consistent with the results on ImageNet. Similarly, SAF and MESA outperform SAM on large-scale models—Wide Resnet and PyradmidNets.

## 4.3 Discussion

**Memory Usage** We evaluate the additional memory usage of SAF and MESA on the ImageNet dataset with the ResNet-50 model. We only present the results on the ImageNet in Table 2, because the memory usage of SAF on the CIFAR dataset is negligible (57 Mb). MESA saves 99.3% memory usage compared to SAF, which allows MESA to be deployed on extremely large-scale datasets.

**Convergence Rate** Intuitively, SAF minimizes the trajectory loss to control the rate of training loss change to obtain flat minima. A critical problem of SAF may be SAF's influence on the convergence rate. We empirically show that the change of the sharpness-inducing term in SAF and MESA (compared to SAM) will not affect the convergence rate during training. Figure 4a illustrates the change of the training loss in each epoch of SGD, SAM, SAF, and MESA. It shows that SAF and MESA converge at the same rate as SGD and SAM.

Table 2: The additional memory used by SAF and MESA on the ImageNet dataset.

| Algorithms | Extra Memory Usage |
|---|---|
| SAF | 14,643 MB |
| MESA | 98 MB |

**Sharpness** We empirically demonstrate that the trajectory loss can be a reliable proxy of the sharpness loss proposed by SAM. We plot the sharpness, which is the measurement of SAM in Equation 6, in each epoch of SAM, SGD, SAF, and MESA during training. As shown in Figure 4b, SAF and MESA minimize the sharpness as SAM does throughout the entire training procedure. Both SAF's and MESA's sharpness decrease significantly at epoch 5, from which the trajectory loss starts to be minimized. SAM's sharpness is lower than SAF's and MESA's in the second half of the training. A plausible reason is that SAM minimizes the sharpness directly. However, SAF and MESA outperform SAM in terms of the test accuracies, as demonstrated in Table 1.

**Visualization of Loss Landscapes** We also demonstrate that minimizing the trajectory loss is an effective way to obtain flat minima. We visualize the loss landscape of converged minima trained by SGD, SAM, SAF, and MESA in Figures 2 and 5. We follow the method to do the plotting proposed by [17], which has also been used in [2, 6]. The $x$- and $y$-axes represent two random sampled orthogonal Gaussian perturbations. We sampled $100 \times 100$ points with random Gaussian perturbations for visualization. The visualized loss landscape clearly demonstrate that SAF can converge to a region as flat as SAM does.

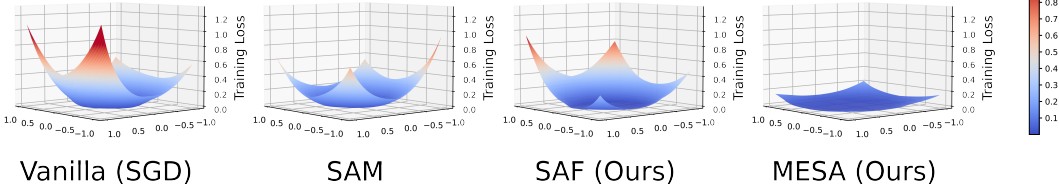

Vanilla (SGD)   SAM   SAF (Ours)   MESA (Ours)

Figure 5: Loss landscapes visualization of the Wide ResNet-28-10 model on the CIFAR-100 dataset trained with SGD, SAM, our proposed SAF and MESA.

Table 3: Classification accuracies and training speed on the CIFAR-10 and CIFAR-100 datasets. The numbers in parentheses (·) indicate the ratio of the training speed w.r.t. the vanilla base optimizer's (SGD's) speed. Green indicates improvement compared to SAM, whereas red suggests a degradation.

| | CIFAR-10 | | CIFAR-100 | |
|---|---|---|---|---|
| **ResNet-18** | Accuracy | images/s | Accuracy | images/s |
| Vanilla (SGD) | $95.61_{\pm 0.02}$ | 3,289 (100%) | $78.32_{\pm 0.02}$ | 3,314 (100%) |
| SAM [8] | $\mathbf{96.50}_{\pm 0.08}$ | 1,657 (50.4%) | $\mathbf{80.18}_{\pm 0.08}$ | 1,690 (51.0%) |
| SAF (Ours) | $96.37_{\pm 0.02}$ | 3,213 (97.6%) | $80.06_{\pm 0.05}$ | 3,248 (98.0%) |
| MESA (Ours) | $96.24_{\pm 0.02}$ | 2,780 (84.5%) | $79.79_{\pm 0.09}$ | 2,793 (84.3%) |
| **ResNet-101** | Accuracy | images/s | Accuracy | images/s |
| Vanilla (SGD) | $96.52_{\pm 0.04}$ | 501 (100%) | $80.68_{\pm 0.16}$ | 501 (100%) |
| SAM [8] | $\mathbf{97.01}_{\pm 0.32}$ | 246 (49.1%) | $\mathbf{82.99}_{\pm 0.04}$ | 249 (49.7%) |
| SAF (Ours) | $96.93_{\pm 0.05}$ | 497 (99.2%) | $82.84_{\pm 0.19}$ | 497 (99.2%) |
| MESA (Ours) | $96.90_{\pm 0.23}$ | 425 (84.8%) | $82.51_{\pm 0.27}$ | 426 (85.0%) |
| **Wide-28-10** | Accuracy | images/s | Accuracy | images/s |
| Vanilla (SGD) | $96.50_{\pm 0.05}$ | 732 (100%) | $81.67_{\pm 0.18}$ | 739 (100%) |
| SAM [8] | $97.07_{\pm 0.11}$ | 367 (50.1%) | $83.51_{\pm 0.04}$ | 370 (50.0%) |
| SAF (Ours) | $97.08_{\pm 0.15}$ | 727 (99.3%) | $\mathbf{83.81}_{\pm 0.04}$ | 729 (98.6%) |
| MESA (Ours) | $\mathbf{97.16}_{\pm 0.23}$ | 617 (84.3%) | $83.59_{\pm 0.24}$ | 625 (84.6%) |
| **PyramidNet-110** | Accuracy | images/s | Accuracy | images/s |
| Vanilla (SGD) | $96.66_{\pm 0.09}$ | 394 (100%) | $81.94_{\pm 0.06}$ | 401 (100%) |
| SAM [8] | $97.25_{\pm 0.15}$ | 194 (49.3%) | $84.61_{\pm 0.06}$ | 198 (49.4%) |
| SAF (Ours) | $97.34_{\pm 0.06}$ | 391 (99.2%) | $84.71_{\pm 0.01}$ | 397 (99.0%) |
| MESA (Ours) | $\mathbf{97.46}_{\pm 0.09}$ | 332 (84.3%) | $\mathbf{84.73}_{\pm 0.14}$ | 339 (84.5%) |

## 5   Other Related Works

The first work that revealed the relation between the generalization ability and the geometry of the loss landscape (sharpness) can be traced back to [12]. Following that, many studies verified the relation between the flat minima and the generalization error [4, 7, 13, 14, 17, 19, 23] . Specifically, Keskar *et al.* [14] proposed a sharpness measure and indicated the negative correlation between the sharpness measure and the generalization abilities. Dinh *et al.* [4] further argued that the sharpness measure can be related to the spectrum of the Hessian, whose eigenvalues encode the curvature information of the loss landscape. Jiang *et al.* [13] demonstrated that one of the sharpness-based measures is the most correlated one among 40 complexity measures by a large-scale empirical study.

SAM [8] solved the sharp minima problem by modifying training schemes to approximate and minimize a certain sharpness measure. The concurrent works [30, 33] propose a model to adversarially perturb trained weights to bias the convergence. The aforementioned methods lead the way for sharpness-aware training despite the computational overhead being doubled over that of the base optimizer. Subsequently, LookSAM [20] and Efficient SAM (ESAM) [6] were proposed to alleviate the computational issue of SAM. Apart from efficiency issues, Surrogate Gap Guided Sharpness-Aware Minimization (GSAM) [36] was proposed to further improve SAM's performance. GSAM places more emphasis on the gradients that minimize the sharpness loss to achieve the best generalization performance among all the research that followed SAM.

# 6 Conclusion and Future Research

In this work, we introduce a novel trajectory loss as an equivalent measure of sharpness, which requires almost no additional computational overhead and preserves the superior performance of sharpness-aware training [6, 8, 20, 36]. In addition to deriving a novel trajectory loss that penalizes sharp and sudden drops in the objective function, we propose a novel sharpness-aware algorithm SAF, which achieves impressive performances in terms of its accuracies on the benchmark CIFAR and ImageNet datasets. More importantly, SAF trains DNNs *at the same speed* as non-sharpness-aware training (e.g., SGD). We also propose MESA as a memory-efficient variant of SAF to avail sharpness-aware training on extremely large-scale datasets. In future research, we will further enhance SAF to automatically sense the current state of training and alter the training dynamics accordingly. We will also evaluate SAF in more learning tasks (such as natural language processing) and enhance it to become a general-purpose training strategy.

## Acknowledgments and Disclosure of Funding

This work is support by Joey Tianyi Zhou's A*STAR SERC Central Research Fund (Use-inspired Basic Research) and the Singapore Government's Research, Innovation and Enterprise 2020 Plan (Advanced Manufacturing and Engineering domain) under Grant A18A1b0045.

Vincent Tan acknowledges funding from a Singapore National Research Foundation (NRF) Fellowship (A-0005077-01-00) and Singapore Ministry of Education (MOE) AcRF Tier 1 Grants (A-0009042-01-00 and A-8000189-01-00).

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
