# A Appendix

## A.1 Derivation of Equation 8 and Equation 12

By substituting Equation 5 back into the LHS of Equation 8, we have,

$$
\begin{aligned}
\mathop{\mathbb{E}}_{\theta_i \sim \mathrm{Unif}(\Theta)} \left[ \gamma_i R_{\mathbb{B}_t}(f_{\theta_i}) R_{\mathbb{B}_i}(f_{\theta_i}) \right] &\approx \mathop{\mathbb{E}}_{\theta_i \sim \mathrm{Unif}(\Theta)} \left[ \eta_i \cos(\Phi_i) \| \nabla_{\theta_i} L_{\mathbb{B}_t}(f_{\theta_i}) \| \, \| \nabla_{\theta_i} L_{\mathbb{B}_i}(f_{\theta_i}) \| \right] \\
&= \mathop{\mathbb{E}}_{\theta_i \sim \mathrm{Unif}(\Theta)} \left[ \eta_i \nabla_{\theta_i} L_{\mathbb{B}_t}(f_{\theta_i})^\top \nabla_{\theta_i} L_{\mathbb{B}_i}(f_{\theta_i}) \right] \\
&\approx \mathop{\mathbb{E}}_{\theta_i \sim \mathrm{Unif}(\Theta)} \left[ L_{\mathbb{B}_t}(f_{\theta_i}) - L_{\mathbb{B}_t}(f_{\theta_{i+1}}) \right] \\
&= \frac{1}{t-1} \left[ L_{\mathbb{B}_t}(f_{\theta_1}) - L_{\mathbb{B}_t}(f_{\theta_2}) + \cdots + L_{\mathbb{B}_t}(f_{\theta_{t-1}}) - L_{\mathbb{B}_t}(f_{\theta_t}) \right] \\
&= \frac{1}{t-1} \left[ L_{\mathbb{B}_t}(f_{\theta_1}) - L_{\mathbb{B}_t}(f_{\theta_t}) \right]
\end{aligned}
$$

When we replace the trajectory from $\Theta = \{\theta_2, \theta_3, \ldots, \theta_{t-1}\}$ to $\mathbb{W}_{\mathrm{EMA}} = \{w_2, w_3, \ldots, w_{t-1}\}$, where $w_i = w_{i-1} - \beta^{t-i} \eta \nabla_{\theta_i} L_{\mathbb{B}_i}(f_{\theta_i})$, $w_1 = v_t$, and $w_t = \theta_t$, and substitute it back into Equation 8, we have

$$
\begin{aligned}
\frac{1}{t-1} \left[ L_{\mathbb{B}_t}(f_{v_t}) - L_{\mathbb{B}_t}(f_{\theta_t}) \right] &= \mathop{\mathbb{E}}_{w_i \sim \mathrm{Unif}(\mathbb{W}_{\mathrm{EMA}})} \left[ L_{\mathbb{B}_t}(f_{w_i}) - L_{\mathbb{B}_t}(f_{w_{i+1}}) \right] \\
&\approx \mathop{\mathbb{E}}_{w_i \sim \mathrm{Unif}(\mathbb{W}_{\mathrm{EMA}})} \left[ \beta^{t-i} \eta_i \nabla_{w_i} L_{\mathbb{B}_t}(f_{w_i})^\top \nabla_{\theta_i} L_{\mathbb{B}_t}(f_{\theta_i}) \right] \\
&= \mathop{\mathbb{E}}_{w_i \sim \mathrm{Unif}(\mathbb{W}_{\mathrm{EMA}})} \left[ \beta^{t-i} \eta_i \cos(\Phi_i) \| \nabla_{w_i} L_{\mathbb{B}_t}(f_{w_i}) \| \, \| \nabla_{\theta_i} L_{\mathbb{B}_t}(f_{\theta_i}) \| \right]. \\
&\approx \mathop{\mathbb{E}}_{w_i \sim \mathrm{Unif}(\mathbb{W}_{\mathrm{EMA}})} \left[ \beta^{t-i} \gamma_i R_{\mathbb{B}_t}(f_{w_i}) R_{\mathbb{B}_t}(f_{\theta_i}) \right].
\end{aligned}
$$

We see that the difference between this and Equation 8 is the additional coefficient $\beta^{t-i}$ of each term. The coefficient $\beta^{t-i}$ is monotonic increasing as $i$ increases, since $0 < \beta < 1$. Therefore, $\beta^{t-i}$ will put *more emphasis* on the *more recent* weights.

## A.2 Derivation of Equation 11

By definition of Exponential Moving Average (EMA), the EMA weights $v_t$ in the $t$-th iterations are as follows

$$
\begin{aligned}
v_t &= \beta v_{t-1} + (1-\beta)\theta_t \\
&= \beta(\beta v_{t-2} + (1-\beta)\theta_{t-1}) + (1-\beta)\theta_t \\
&= \vdots \\
&= \beta^{t-1} v_1 + (1-\beta)(\theta_t + \beta\theta_{t-1} + \beta^2\theta_{t-2} + \cdots + \beta^{t-2}\theta_2).
\end{aligned} \tag{14}
$$

Let $g_i = \eta \nabla_{\theta_i} L_{\mathbb{B}_i}(f_{\theta_i})$. We recall that

$$
\theta_t = \theta_{t-1} - g_{t-1} = \theta_{t-2} - g_{t-2} - g_{t-1} = \ldots = \theta_1 - \sum_{i=1}^{t-1} g_i \tag{15}
$$

Substituting Equation 15 into Equation 14, we obtain

$$
v_t = \beta^{t-1}v_1 + (1-\beta)\left(\theta_1 - \sum_{i=1}^{t-1}g_i + \beta\left(\theta_1 - \sum_{i=1}^{t-2}g_i\right) + \ldots + \beta^{t-2}(\theta_1 - g_1)\right)
$$

$$
= \beta^{t-1}v_1 + (1-\beta)\left(\frac{1-\beta^{t-1}}{1-\beta}\theta_1 - \sum_{i=1}^{t-1}\frac{1-\beta^{t-i}}{1-\beta}g_i\right)
$$

$$
= \beta^{t-1}v_1 + (1-\beta^{t-1})\theta_1 - \sum_{i=1}^{t-1}(1-\beta^{t-i})g_i
$$

$$
= \theta_1 - \sum_{i=1}^{t-1}(1-\beta^{t-i})g_i
$$

$$
= \theta_t + \sum_{i=1}^{t-1}\beta^{t-i}g_i = \theta_t + \sum_{i=1}^{t-1}\beta^{t-i}\eta\nabla_{\theta_i}L_{\mathbb{B}_i}(f_{\theta_i}),
$$

as desired.

### A.3 Training Details

We search the training parameters of SGD, SAM, SAF and MESA via grid searches on an isolated validation set of CIFAR datasets. The learning rate is chosen from the set $\{0.01, 0.05, 0.1, 0.2\}$, the weight decay from the set $\{5 \times 10^{-4}, 1 \times 10^{-3}\}$, and the batch size from the set $\{64, 128, 256\}$. We choose the hyperparameters that achieves the best test accuracies and report them in Table 4. We also report the hyperparameters on the ImageNet datasets in Table 5.

The set of hyperparameters $\{E_{\text{start}}, \tilde{E}, \tau\}$ is consistent among the experiments on the CIFAR and ImageNet datasets. We set $\{E_{\text{start}} = 5, \tilde{E} = 3, \tau = 5\}$ for each model on the CIFAR-10/100 dataset. For the ImageNet dataset, we set $\{E_{\text{start}} = 10, \tilde{E} = 3, \tau = 5\}$ for the ResNets and $\{E_{\text{start}} = 150, \tilde{E} = 3, \tau = 5\}$ for the ViT.

Table 4: Hyperparameters for training from scratch on CIFAR10 and CIFAR100

| **ResNet-18** | **CIFAR-10** | | | | **CIFAR-100** | | | |
|---|---|---|---|---|---|---|---|---|
| | SGD | SAM | SAF | MESA | SGD | SAM | SAF | MESA |
| Epoch | 200 | | | | 200 | | | |
| Batch size | 128 | | | | 128 | | | |
| Data augmentation | Basic | | | | Basic | | | |
| Peak learning rate | 0.05 | | | | 0.05 | | | |
| Learning rate decay | Cosine | | | | Cosine | | | |
| Weight decay | $5 \times 10^{-4}$ | | | | $5 \times 10^{-4}$ | | | |
| $\rho$ | - | 0.05 | - | - | - | 0.05 | - | - |
| $\beta$ | | 0.9995 | | | | 0.9995 | | |
| $\lambda$ | - | - | 0.3 | 0.8 | - | - | 0.3 | 0.8 |
| **ResNet-101** | SGD | SAM | SAF | MESA | SGD | SAM | SAF | MESA |
| Epoch | 200 | | | | 200 | | | |
| Batch size | 128 | | | | 128 | | | |
| Data augmentation | Basic | | | | Basic | | | |
| Peak learning rate | 0.05 | | | | 0.05 | | | |
| Learning rate decay | Cosine | | | | Cosine | | | |
| Weight decay | $5 \times 10^{-4}$ | | | | $5 \times 10^{-4}$ | | | |
| $\rho$ | - | 0.05 | - | - | - | 0.05 | - | - |
| $\beta$ | | 0.9995 | | | | 0.9995 | | |
| $\lambda$ | - | - | 0.3 | 0.8 | - | - | 0.3 | 0.8 |
| **Wide-28-10** | SGD | SAM | SAF | MESA | SGD | SAM | SAF | MESA |
| Epoch | 200 | | | | 200 | | | |
| Batch size | 128 | | | | 128 | | | |
| Data augmentation | Basic | | | | Basic | | | |
| Peak learning rate | 0.05 | | | | 0.05 | | | |
| Learning rate decay | Cosine | | | | Cosine | | | |
| Weight decay | $1 \times 10^{-3}$ | | | | $1 \times 10^{-3}$ | | | |
| $\rho$ | - | 0.1 | - | - | - | 0.1 | - | - |
| $\beta$ | | 0.9995 | | | | 0.9995 | | |
| $\lambda$ | - | - | 0.3 | 0.8 | - | - | 0.3 | 0.8 |
| **PyramidNet-110** | SGD | SAM | SAF | MESA | SGD | SAM | SAF | MESA |
| Epoch | 200 | | | | 200 | | | |
| Batch size | 128 | | | | 128 | | | |
| Data augmentation | Basic | | | | Basic | | | |
| Peak learning rate | 0.05 | | | | 0.05 | | | |
| Learning rate decay | Cosine | | | | Cosine | | | |
| Weight decay | $5 \times 10^{-4}$ | | | | $5 \times 10^{-4}$ | | | |
| $\rho$ | - | 0.2 | - | - | - | 0.2 | - | - |
| $\beta$ | | 0.9995 | | | | 0.9995 | | |
| $\lambda$ | - | - | 0.3 | 0.8 | - | - | 0.3 | 0.8 |

Table 5: Hyperparameters for training from scratch on ImageNet

| ImageNet | ResNet-50 | | | | ResNet-101 | | | |
|---|---|---|---|---|---|---|---|---|
| | SGD | SAM | SAF | MESA | SGD | SAM | SAF | MESA |
| Epoch | 90 | | | | 90 | | | |
| Batch size | 4096 | | | | 4096 | | | |
| Data augmentation | Inception-style | | | | Inception-style | | | |
| Peak learning rate | 1.4 | | | | 1.4 | | | |
| Learning rate decay | Cosine | | | | Cosine | | | |
| Weight decay | $3 \times 10^{-5}$ | | $1 \times 10^{-4}$ | | $3 \times 10^{-5}$ | | | |
| $\rho$ | - | 0.05 | - | - | - | 0.05 | - | - |
| $\beta$ | 0.9998 | | | | 0.9998 | | | |
| $\lambda$ | - | - | 0.3 | 0.3 | - | - | 0.3 | 0.3 |

| ImageNet | ResNet-152 | | | | ViT-S/32 | | | |
|---|---|---|---|---|---|---|---|---|
| | SGD | SAM | SAF | MESA | SGD | SAM | SAF | MESA |
| Epoch | 90 | | | | 300 | | | |
| Batch size | 4096 | | | | 4096 | | | |
| Data augmentation | Inception-style | | | | Inception-style | | | |
| Peak learning rate | 1.4 | | | | $3 \times 10^{-3}$ | | | |
| Learning rate decay | Cosine | | | | Cosine | | | |
| Weight decay | $3 \times 10^{-5}$ | | | | 0.3 | | | |
| $\rho$ | - | 0.05 | - | - | - | 0.05 | - | - |
| $\beta$ | 0.9998 | | | | 0.9998 | | | |
| $\lambda$ | - | - | 0.3 | 0.3 | - | - | 0.3 | 0.3 |