# OpenReview forum: "Sharpness-Aware Training for Free"
_NeurIPS.cc/2022/Conference — NeurIPS 2022 Accept_

### Official Review · Reviewer_3mAW · 2022-06-14

**Rating:** 4
**Confidence:** 2
**Soundness:** 2 fair
**Presentation:** 2 fair
**Contribution:** 2 fair

**Summary:**

This paper proposes two methods for sharpness aware minimization (SAM) without the cost of SAM, as SAM doubles training cost. The methods are 1) SAF which adds a distillation loss from the predictions of prior epochs and 2) MESA which adds a distillation loss from the exponential moving average of the weights. For resnets on imagenet, SAF and MESA match or outperform other methods including SAM, ESAM, and Vanilla SGD.

**Questions:**

I understand that there is not space to answer all of these questions.
- Why is the ResNet baseline for SAM much lower than that reported in the SAM paper?
- Why is MESA only 15% more expensive than standard training
- The best ResNet50 in the paper is < 78%. However, modern day ResNets (e.g., from https://arxiv.org/pdf/2110.00476.pdf in the popular timm library, which is used in this paper) can get ~80%. Are the methods in this paper additive with more modern approaches.
- Why is GSAM much better than all other methods for the ViT model (minor).

**Limitations:**

The paper claims to address this in Section 3.3 but a revision could benefit from an explicit limitations section (minor).

**Strengths And Weaknesses:**

Strengths:
- The paper tackles and interesting and practical problem, which is how to make training better without the added cost of methods like SAM.
- For ResNets on ImageNet, the empirical performance when compared to baselines is good. Moreover, this is without the cost.

Weaknesses:
- I do not understand how MESA is only 15% more expensive than standard training. If I am understanding correctly, MESA requires a forward pass through both the EMA weights and the standard weights. Doesn't this require two forward passes where there was previously only one?
- It seems like free should not be in the name of SAF because there is an additional memory overhead (minor)
- I did not understand very well how SAF and MESA relate to SAM. In particular, Equation 8 is not well motivated and I did not see where the KL came from.
- I have a few concerns with the ResNet baselines: 1) The ResNet baseline for SAM is below that reported in the SAM paper -- could this perhaps be because perturbations are synced? 2) Modern ResNets, e.g. with techniques from the timm library which is the library used in this paper can get to 80% accuracy (https://arxiv.org/pdf/2110.00476.pdf). Are the methods presented in this paper additive with these modern approaches?
-

---

> ### Author Response · Authors · 2022-08-02
> **Response**
>
>
> Thank you for your  feedback. Our responses to the weak points and questions are as below.
>
> ---
> **Q1:** Why is the ResNet baseline for SAM much lower than that reported in the SAM paper? Are the methods in this paper additive with more modern approaches.
>
>
> **A1:** This is because of the  difference in the setting. The recent follow-up works for SAM adopt a different setting from the original SAM work. As we stated on Line 230, we followed the experimental settings of SAM’s follow-up works [2,21,6,37] for fair comparison with them. Our reported results of SAM and SAM’s variants **exactly** match the results in [2,21,6,37] (and the experimental results in [2,21,6,37] are all the same).
>
> Details on the settings: SAM uses 100 epochs (vs 90 epochs in our setting) and SAM uses label smoothing of 0.1 (vs no label smoothing in our setting). Thus SAM reports 77.1% accuracy on ResNet 50 SGD while [2,21,6,37] and our paper reports 76.0% accuracy.
>
> For comparison to the original SAM work, we conducted experiments to use stronger data augmentation (Mix Up and Cut Mix only) for our proposed MESA. Our SAF achieves a consistent 1.2% accuracy improvement over the reported results of the original SAM work. The results are listed below.
>
> |ResNet-50|SAM Basic data augmentation | Augmentation with Mixup and Cut Mix |
> | ----------- | :-----------: |:-----------:|
> |SAM|76.0|77.5|
> |SAF|77.5|78.7|
>
>
> MESA can still be effective for the training with stronger data augmentation. The architecture of ResNet 50 model in https://arxiv.org/pdf/2110.00476.pdf has been improved and trained with all data augmentation strategies to achieve 80% accuracy. We will evaluate SAF and MESA with the improved ResNets in a revised version of the paper. As we stated in footnote 2 of page 7, we failed to reproduce GSAM on the ImageNet by modifying the example on the CIFAR 10 dataset.
>
> ---
> **Q2:** Why is MESA only 15% more expensive than standard training
>
>  **A2:** MESA only requires one more forward pass, which requires **no computation of any gradients**, to construct the trajectory loss (line 9 in Alg 1). The forward propagation is conducted under “with torch.no_grad()” in the pytorch framework, which only requires 15% more computations than the standard forward and backward propagation. SAF does not have such an additional forward pass as SAF loads the saved outputs (line 5 in Alg 1). We will emphasize this point when we discuss MESA in the revised version of the paper.
>
> ---
>
> **Q3:** Why is GSAM much better than all other methods for the ViT model (minor)
> **A3:** This is possibly because of the data augmentation and tricks such as Random Erase [1]. The authors of GSAM did not release their codes for their experiments on the ImageNet dataset. +footnote 2
>
> [1] Zhong, Zhun, et al. "Random erasing data augmentation." Proceedings of the AAAI conference on artificial intelligence. Vol. 34. No. 07. 2020.
>
> ---
> **Q4:** I did not understand very well how SAF and MESA relate to SAM. In particular, Equation 8 is not well motivated and I did not see where the KL came from.
>
> **A4:**  Our methodology section details how we derive SAF and MESA from SAM to address its efficiency issue. We aim to derive an equivalent term as the sharpness term defined by SAM in Equation (5). The equivalent term should require much fewer computations than SAM. Equation (7) shows that the loss difference term is precisely the equivalent term we are looking for. As we stated in Line 175, the vanilla loss will be canceled out if we minimize the equivalent term in Equation (7) directly. Thus, we propose the trajectory loss in Equation (8) for SAF, which is derived from Equation (7). The reason for using KL is because minimizing the CE loss is equivalent to minimizing the KL loss. The proof is delineated below.
>
> We are given that $\\{\hat{y_i}\\}\_{i=1}^n$ are the ground truths and $\\{f_\theta(x_i)\\}_{i=1}^n$ are the outputs of the neural network $f_\theta$.  The definition of the KL loss is as follow,
>
> $\mathrm{KL}(\hat{y},f_\theta(x))  =\sum\limits_{i=1}^n \hat{y_i} \log\hat{y_i} - \hat{y_i} \log f_\theta(x_i)$
>
> The definition of the Cross-Entropy loss is as follow,
>
> $\mathrm{CE}(\hat{y},f_\theta(x))= \sum\limits_{i=1}^n- \hat{y_i} \log f_\theta(x_i) =\mathrm{KL}(\hat{y},f_\theta(x)) - \sum\limits_{i=1}^n \hat{y_i} \log\hat{y_i}$
>
> Because the term $H(\hat{y})=\sum\limits_{i=1}^n \hat{y_i} \log\hat{y_i}$ is a constant with respect to the variable $\theta$, we have,
>
> $\underset{\theta}{\arg\min} \mathrm{KL}(\hat{y},f_\theta(x)) = \underset{\theta}{\arg\min} [\mathrm{CE}(\hat{y},f_\theta(x))+H(\hat{y})]= \underset{\theta}{\arg\min}\mathrm{CE}(\hat{y},f_\theta(x))$
>
> Therefore, we have proven that minimizing the CE loss is equivalent to minimizing the KL loss.

---

> > ### Comment · Reviewer_3mAW · 2022-08-04
> > **Thanks for the response.**
> >
> > Thank you for the response. I am surprised that an inference forward pass is only 15% of the time for a standard forward and backward pass so would recommend double checking this. I also appreciate the experiments with additional data augmentation, though the models are still far from SoTA despite the SoTA recipes being public. Despite these concerns, I am changing my recommendation to accept but I am not confident in this recommendation.

---

> > > ### Author Response · Authors · 2022-08-05
> > > **Thanks for the comments and raising the score**
> > >
> > > Dear Reviewer 3mAW,
> > > Thanks for the comments and for raising the score of our paper.
> > > Best regards,
> > > The authors

---

### Official Review · Reviewer_nBtT · 2022-07-08

**Rating:** 6
**Confidence:** 4
**Soundness:** 4 excellent
**Presentation:** 4 excellent
**Contribution:** 3 good

**Summary:**

This paper focuses on the computation cost perspective of Sharpness-Aware Optimization (SAM), which has 2x cost compared with SGD.
The first proposed algorithm, SAM for Free (SAF), derives an alternative/surrogate optimization objective which can uses the past optimization trajectory to estimate the sharpness loss. The SAF algorithm has almost the same computation cost as SGD, while inducing some extra memory usage. To mitigate the memory cost, the authors propose the second algorithm, Memory-Efficient Sharpness-Aware Training (MESA), which reduces the memory usage significantly at the tradeoff of 15% more computation cost. The empirical results of SAF and MESA are pretty good compared with SAM and SGD on popular benchmarks (CIFAR and ImageNet), indicating that the the proposed algorithms indeed reduce the computation costs of SAM while maintaining the performance.

**Questions:**

1. Is the reduction of computation cost from 2x to 1x SGD really meaningful?
Theoretically speaking, 1x and 2x costs are about the same. Empirically, I know there are many factors that affecting the computation cost of training, and I'm not sure if 50% cost reduction is very meaningful practically.

2.  See Weakness-1 above. That's one of my questions.

**Limitations:**

See the Questions section above.

**Strengths And Weaknesses:**

Strengths:

1. The SAF algorithm is proposed in a principled way: The authors studied the SAM loss carefully, and obtained some interesting observations (Sec. 3.1), naturally leading to an elegant algorithm

2. The authors indeed make the algorithm practical: Even though the general idea (Sec. 3.1) is elegant, it may still have some limitations when being used as a practical optimizer. The authors tackled some numerical challenge in Sec. 3.2, and moreover, they mitigated the memory cost issues by using the trick of exponential moving average (EMA), leading to the memory-efficient version (MESA). I really appreciate the efforts the authors spent on making the algorithm practically useful.

3. The empirical results are good: The proposed algorithms, SAF and MESA, have slightly better performance than SAM with ~1/2 computation cost. Also, the extra memory cost of MESA is indeed negligible compared with SAF.

Weakness:

1. Is the replacement of cross-entropy with KL (in Sec. 3.2) really necessary? I think cross-entropy is still the best choice for classification, so I guess the replacement may have some negative impacts. I think it's better to have an ablation study on this replacement.

---

> ### Author Response · Authors · 2022-08-02
> **Response**
>
> We appreciate your valuable comments and answer your questions in order.
>
> ---
> **Q1:** Is the replacement of cross-entropy with KL (in Sec. 3.2) really necessary?
>
> **A1:** Minimizing the KL loss is equivalent to minimizing the cross-entropy loss, which is proved below.
>
>
> We are given that $\\{\hat{y_i}\\}\_{i=1}^n$ are the ground truths and $\\{f_\theta(x_i)\\}_{i=1}^n$ are the outputs of the neural network $f_\theta$.  The definition of the KL loss is as follow
>
> $\mathrm{KL}(\hat{y},f_\theta(x))  =\sum\limits_{i=1}^n \hat{y_i} \log\hat{y_i} - \hat{y_i} \log f_\theta(x_i)$
>
> The definition of the Cross-Entropy loss is as follow,
>
> $\mathrm{CE}(\hat{y},f_\theta(x))= \sum\limits_{i=1}^n- \hat{y_i} \log f_\theta(x_i) =\mathrm{KL}(\hat{y},f_\theta(x)) - \sum\limits_{i=1}^n \hat{y_i} \log\hat{y_i}$
>
> Because the term $H(\hat{y})=\sum\limits_{i=1}^n \hat{y_i} \log\hat{y_i}$ is a constant with respect to the variable $\theta$, we have,
>
> $\underset{\theta}{\arg\min} \mathrm{KL}(\hat{y},f_\theta(x)) = \underset{\theta}{\arg\min} [\mathrm{CE}(\hat{y},f_\theta(x))+H(\hat{y})]= \underset{\theta}{\arg\min}\mathrm{CE}(\hat{y},f_\theta(x))$
>
>
> ---
> **Q2:** Is the reduction of computation cost from 2x to 1x SGD really meaningful?
>
> **A2:** Yes, it is extremely meaningful especially for training large-scale neural networks and the commercialization of deep learning models.  For example, if one trains a ResNet 152 model on ImageNet dataset with 8 Nvidia V-100 GPUs for 3 days, considering ​​Google Cloud Platform charges $2.28 per GPU per hour [Refer](https://venturebeat.com/2021/10/15/ai-weekly-ai-model-training-costs-on-the-rise-highlighting-need-for-new-solutions/)), the total cost of SAM will be \\$2,646 (\\$2.28 $\times$ 8 GPUs $\times$ 2 $\times$ 3 days $\times$ 24 hours) , while the total cost of SAF will be $1,313. There is a significant cost savings between implementing SAM and SAF.
>
> Thus, significant efforts over the past couple of years have been devoted to halving the computational resources required for SAM (to match the base optimizer SGD), and our work shows that we can obtain the generalization ability of SAM without suffering from the curse of double computation.

---

> > ### Comment · Reviewer_nBtT · 2022-08-02
> > **Response to authors**
> >
> > Thank you for the clarifications.

---

### Official Review · Reviewer_RMT8 · 2022-07-10

**Rating:** 3
**Confidence:** 4
**Soundness:** 1 poor
**Presentation:** 2 fair
**Contribution:** 2 fair

**Summary:**

This paper proposes efficient implementations of sharpness-aware training. Specifically, this paper proposed Sharpness-Aware Training for Free (SAF) which requires cached prediction for previous checkpoints, and Memory-Efficient Sharpness-Aware (MESA) training which requires two forward propagation and one back propagation for each optimization step to achieve sharpness-aware training. While these approaches require relatively light computation costs compared to the original SAM, they achieved competitive accuracy in various tasks including CIFAR-10/100 and ImageNet-1k.

**Questions:**

Q1. In Fig. 2 and Fig. 5, the authors provide visualization of the loss landscape for various sharpness-aware training. Although the results showed that SAF/MESA found flatter local minima than SAM, one cannot find the clear intuition behind these empirical results. Also, while these visualizations tell us that MESA found flatter local minima than SAF for WRN28-10/CIFAR-100, the test accuracy of SAF is better than MESA. Therefore, it can be interpreted as flatter local minima considered in this paper would not result in better test accuracy in practice. Authors should clarify this paradox to remove the confusion of readers.

Q2. As mentioned in the original SAM paper, the SAM optimizer can find local minima with better test accuracy with longer training (See Table 2 of [1]). Can SAF and MESA also utilize the longer training without overfitting? I think this property would be important to the scalability of SAF & MESA.

**Limitations:**

While this paper provides a novel interpretation of sharpness by leveraging the trajectory. several points are needed to improve the clarity and significance of the paper.

1. It would be recommended to provide the detailed derivations of L163-L167. While the intuition behind these lines is clear, further clarity can be attained by providing the entire derivation of equations to readers.

2. It should be clarified why SAM accuracy in Table 1 is severely lower than the original SAM paper [1]. Since they did not report any confidence interval in Table 1, this accuracy gap can change the interpretation of experimental results in Section 4.1.

3. It would be recommended to add the results of SAF/MESA for longer training and stronger augmentation. Since many SOTA models in computer vision [3,4] exploit longer training and stronger augmentation, the scalability of SAF/MESA should be validated with those settings.

[3] https://pytorch.org/blog/how-to-train-state-of-the-art-models-using-torchvision-latest-primitives

[4] Liu, Zhuang, et al. "A convnet for the 2020s." Proceedings of the IEEE/CVF Conference on Computer Vision and Pattern Recognition. 2022.

** After discussion phase **

Thanks to the clarification of the authors, I resolved my misunderstanding of this paper. However, I think these discussion points should be reflected in the main paper: Their key trick using $\arg \min$ is not specified in their derivation of the main paper. Also, further clarification on differences between main references (e.g., label smoothing, number of epochs) should be clarified in their main paper since the gap between their performances is sufficiently large to change the interpretation of experimental results.


**Strengths And Weaknesses:**

* Originality: This paper provides a novel idea to estimate the sharpness by leveraging the trajectory of weights (Eq. 7). Based on this idea, they proposed time-efficient (SAF) and memory-efficient (MESA) implementations of sharpness-aware training [+].

* Clarity: While Fig. 3 is intuitive to understand the key idea of this paper [+], it is still unclear to me that the equation in L164 and the second approximation in eq 7. Also, it is unclear that \Phi_t is defined on what parameter of the optimization step [-].

* Significance: I found that there exists a gap between reported SAM accuracies in previous studies and this paper. In Table 1 of this paper, the authors reported SAM accuracy as 76.9/78.6/79.3 for ResNet-50/101/152. However, the original SAM paper reported their accuracy as 77.5/79.8/80.8 with 100 epoch training (See Table 2 of [1]). I think this gap is critical since the accuracy of SAF 77.8/79.3/80.0, the minor improvements only exist for ResNet-50, and MESA 77.5/79.1/80.0, there are no improvements for SAM. Also, the ViT-S/32 results in Table 1 also make some confusion: In Table 2 of [2], they reported the accuracy of ViT-S/32-SAM as 70.5, which is severely better performing than the 68.9 in this paper. The authors should clarify this for a fair comparison.

[1] Foret, Pierre, et al. "Sharpness-aware minimization for efficiently improving generalization." arXiv preprint arXiv:2010.01412 (2020).

[2] Chen, Xiangning, Cho-Jui Hsieh, and Boqing Gong. "When vision transformers outperform ResNets without pre-training or strong data augmentations." arXiv preprint arXiv:2106.01548 (2021).

---

> ### Author Response · Authors · 2022-08-02
> **Response (1/3)**
>
> Thank you for the comments and suggestions! We answer your questions below.
>
>
> ---
>
> **Q1:** I found that there exists a gap between reported SAM accuracies in previous studies and this paper.
>
> **A1:** This is because of the  differences in the settings. The recent follow-up works for SAM adopt different settings from the original SAM work. As we stated on Line 230, we followed the experimental settings of SAM’s follow-up works [2,21,6,37] for fair comparison with them. Our reported results of SAM and SAM’s variants **exactly** match the results in [2,21,6,37] (and the experimental results in [2,21,6,37] are all the same).
>
> More specifically, SAM uses 100 epochs (vs 90 epochs in our setting) and SAM uses label smoothing of 0.1 (vs no label smoothing in our setting). Thus SAM reports 77.1% accuracy with  ResNet 50 SGD while [2,21,6,37] and our paper report 76.0% accuracy. The mentioned paper  [2]  (cited as [2] in our submission) is the first work to use such different settings than SAM.
>
> To further clarify this point, we conducted experiments with stronger data augmentation (Mix Up and Cut Mix only) for our MESA. Our SAF achieves  1.2% accuracy improvement over the reported results of the original SAM work. The results are listed below.
>
>
> |ResNet-50|basic data augmentation | strong data augmentation |
> | ----------- | :-----------: |:-----------:|
> |SAM|76.0 [2,21,6,37] |77.5 (SAM reported)|
> |MESA|77.5|78.7|
>
>
> The discrepancy between the reported results for ViT-S/32 in [2] is also due to the differences in the ettings. The ViT paper reports ViT results with strong data augmentations while [2] reports ViT results without the augmentations of Mix up and RandAugment. We carefully reproduced the ViT-SAM results using only basic data augmentation (inception style) and only obtained 68.9% accuracy for ViT S/32, which **exactly** matches the reported results in LookSAM paper [21].
>
> We also conducted experiments to use stronger data augmentation (Cut Mix and Random Erase) for our SAF. Our MESA achieves a 1.7% accuracy improvement over the reported results of [2]. The results are listed below.
>
> |ViT-S/32| basic data augmentation |strong data augmentation |
> | ----------- | :-----------: |:-----------:|
> |ViT|68.9 [21] |70.5 [2] |
> |SAF|69.6|72.2|
>
> ---
> **Q2:** It would be recommended to add the results of SAF/MESA for longer training and stronger augmentation.
>
> **A2:** Thanks for the  suggestions. We conducted the experiments on training the ResNet 50  by SAF for longer epochs (90 -> 200 epochs) with the basic data augmentation (only Inception-style augmentation). We cite the 200-epoch SGD results with the same setting in [1] for reference. The results are listed below.
>
>
> |ResNet 50  |90 epochs|200 epochs |
> | ----------- | :-----------: |:-----------:|
> |SGD|76.0|76.4 [1] |
> |SAF|77.8|78.5|
>
> The results indicate that our SAF is still much better (2.1% improve) than SGD in the longer training (200 epochs) setting.
>
> We also conducted experiments to train ResNet 50 by MESA with stronger data agumentation (Mix up and Cut mix). The results are listed below.
>
>
> |ResNet-50|Basic data augmentation | Augmentation with Mixup and Cut Mix |
> | ----------- | :-----------: |:-----------:|
> |SAM|76.0|77.5|
> |SAF|77.5|78.7|
>
> The results show that our MESA is still effective in the stronger data augmentation setting.
>
>
> [1] Kwon, Jungmin, et al. "Asam: Adaptive sharpness-aware minimization for scale-invariant learning of deep neural networks." International Conference on Machine Learning. PMLR, 2021.
>
> [2] Foret, Pierre, et al. "Sharpness-aware Minimization for Efficiently Improving Generalization." International Conference on Learning Representations. 2020.

---

> > ### Author Response · Authors · 2022-08-02
> > **Response (2/3)**
> >
> > **Q3:** It would be recommended to provide the detailed derivations of L163-L167.
> >
> > **A3:**  In our submission, we have already given a detailed derivation of Equation (11)  (Lines 202 to 203) in the supplementary document. This derivation is analogous to the derivation of Equation (7) (Lines 166 to 167). We provide the detailed derivation of Lines 163 to 167 here.
> >
> > Our motivation has been clearly stated in the paper, which is to derive an equivalent term of the sharpness $R_{\mathbb{B}\_t}(f_{\theta_t})$ **without additional computations**. The derivations in Lines 164 and 165 are meant to find such an equivalent term. As defined in Equation (5),  $R_{\mathbb{B}\_t}(f_{\theta_t})$ is always non-negative, we have,
> >
> > $\underset{\theta_t}{\arg\min} R_{\mathbb{B}\_t}(f_{\theta_t})= \underset{\theta_t}{\arg\min} \gamma_t R_{\mathbb{B}\_t}(f_{\theta_t})R_{\mathbb{B}\_t}(f_{\theta_t})$
> >
> > $=\underset{\theta_t}{\arg\min}[ \gamma_t R_{\mathbb{B}\_t}(f_{\theta_t})R_{\mathbb{B}\_t}(f_{\theta_t})+\gamma_i R_{\mathbb{B}\_t}(f_{\theta_i})R_{\mathbb{B}\_i}(f_{\theta_i})]$
> >
> >
> >
> > Note that for $i \ne t$,  $\gamma_i R_{\mathbb{B}\_t}(f_{\theta_i})R_{\mathbb{B}\_i}(f_{\theta_i})]$ is a constant with respect to the variable $\theta_t$. Therefore, we traverse all the terms that satisfy $i \neq t$, and thus we obtain the LHS of Equation (7) in Lines 164 to 165.
> >
> > For Equation (7) (Lines 166 to 167), we intend to demonstrate that $\mathop{\mathbb{E}}_{\theta_i \sim \mathrm{Unif}( \Theta)}[ \gamma_i R_{\mathbb{B}\_t}(f_{\theta_i})R_{\mathbb{B}\_i}(f_{\theta_i})]$ is the equivalent term we are looking for. We have,
> >
> > $\mathop{\mathbb{E}}\_{\theta_i \sim \mathrm{Unif}( \Theta)}[ \gamma_i R_{\mathbb{B}\_t}(f_{\theta_i})R_{\mathbb{B}\_i}(f_{\theta_i})]$
> >
> > $\approx \mathop{\mathbb{E}}\_{\theta_i \sim \mathrm{Unif}( \Theta)} \big[ \eta_i \cos(\Phi_i) \| \nabla_{\theta_i} L_{\mathbb{B}\_t}(f_{\theta_i})\| \, \| \nabla_{\theta_i} L_{\mathbb{B}\_i}(f_{\theta_i}) \| \big]\qquad$ Substitute Equation (5)
> >
> > $= \mathop{\mathbb{E}}\_{\theta_i \sim \mathrm{Unif}( \Theta)} \big[\eta_i \nabla_{\theta_i} L_{\mathbb{B}\_t}(f_{\theta_i})^\top  \nabla_{\theta_i} L_{\mathbb{B}\_i}(f_{\theta_i}) \big]$
> >
> > $\approx \mathop{\mathbb{E}}\_{\theta_i \sim \mathrm{Unif}( \Theta)}\big[ L_{\mathbb{B}\_t}(f_{\theta_i}) - L_{\mathbb{B}\_t}(f_{\theta_{i+1}}) \big]\qquad$ Apply a First-order Taylor Expansion
> >
> > $=\frac{1}{t-1} \big[L_{\mathbb{B}\_t}(f_{\theta_1}) -L_{\mathbb{B}\_t}(f_{\theta_2})+\cdots+L_{\mathbb{B}\_t}(f_{\theta_{t-1}})-L_{\mathbb{B}\_t}(f_{\theta_t})\big]$
> >
> > $= \frac{1}{t-1} \big[ L_{\mathbb{B}\_t}(f_{\theta_1}) -L_{\mathbb{B}\_t}(f_{\theta_t})\big]$
> >
> >
> > Therefore, we have shown that minimizing the loss difference $L_{\mathbb{B}\_t}(f_{\theta_1}) -L_{\mathbb{B}\_t}(f_{\theta_t})$ is approximately equivalent to minimizing the sharpness. We have added this detailed derivation to the revised appendix.
> >
> >
> >
> > ---
> >
> > **Q4:** Although the results showed that SAF/MESA found flatter local minima than SAM, one cannot find the clear intuition behind these empirical results.
> >
> > **A4:** Intuitively, our proposed trajectory loss prevents the model from overfitting to  some critical training samples by suppressing the change of training loss on them, and thus helps the model in seeking a flatter minimum (Figure 3 in our submission). More specifically, for those samples whose predictions (or instance-wise loss) changes fast, minimizing our trajectory loss would suppress their  change in losses  and consequent overfitting by enforcing their predictions to be close to the ones from the past model (SAF) or EMA model (MESA).
> >
> > This motivation is supported by our empirical observations. Some training samples’ predictions are changing significantly between adjacent epochs. For example, about 25% of training samples’ predictions are different from the previous epoch on the CIFAR 100 dataset. Among them, half (12.5%) of training samples’ predictions are changing from correct predictions to the wrong prediction. SAF and MESA can decrease the percentage of these samples from 25% to 21%.

---

> > > ### Author Response · Authors · 2022-08-02
> > > **Response (3/3)**
> > >
> > > **Q5:** Although the results showed that SAF/MESA found flatter local minima than SAM, one cannot find the clear intuition behind these empirical results.
> > >
> > > **A5:** Intuitively, our proposed trajectory loss prevents the model from overfitting to  some critical training samples by suppressing the change of training loss on them, and thus helps the model in seeking a flatter minimum (Figure 3 in our submission). More specifically, for those samples whose predictions (or instance-wise loss) changes fast, minimizing our trajectory loss would suppress their  change in losses  and consequent overfitting by enforcing their predictions to be close to the ones from the past model (SAF) or EMA model (MESA).
> > >
> > > This motivation is supported by our empirical observations. Some training samples’ predictions are changing significantly between adjacent epochs. For example, about 25% of training samples’ predictions are different from the previous epoch on the CIFAR 100 dataset. Among them, half (12.5%) of training samples’ predictions are changing from correct predictions to the wrong prediction. SAF and MESA can decrease the percentage of these samples from 25% to 21%.

---

> > > > ### Comment · Reviewer_RMT8 · 2022-08-06
> > > > **Additional questions - 1**
> > > >
> > > > * **On reproducing SAM**
> > > >
> > > >   While authors argued that previous works [2, 6, 21, 37] exactly match the results of thier table for SAM, I find **this is not true** for [2, 21, 37]:
> > > >
> > > >   * Xiangning et al. [2] reported the performance of ResNet50-SAM with Inception-style preprocessing as 76.7 %(See the first row in Table 2 in [2]), which is 0.7% better than 76.0% in their answer.
> > > >   * Yong et al. [21] **did not report the performance of ResNet50-SAM** with Inception-style preprocessing.
> > > >   * Juntang et al. [37] reported the performance of ResNet50-SAM with Inception-style preprocessing as 76.9 %(See the first row in Table 1 in [37]), which is 0.9% better than 76.0% in their answer.
> > > >
> > > >   Also, the 76.0% in author response does not match to the reported value in their paper (76.9% in Table 1), neither. Also I would like to point out that original SAM paper use basic data augmentation (See page 5 of original SAM: '*In this setting, following prior work (He et al., 2015; Szegedy et al., 2015), we resize and crop images to 224-pixel resolution, normalize them, and use batch size 4096, initial learning rate 1.0, cosine learning rate schedule, SGD optimizer with momentum 0.9, label smoothing of 0.1, and weight decay 0.0001.*') to get 77.5% in their paper (Also, see the open sourced code of SAM: https://github.com/google-research/sam/blob/main/sam_jax/datasets/dataset_source_imagenet.py)
> > > >
> > > >   Also, second Table in the A1 have some typos. Does the first row of this Table mean SAM? If so, then I think author incorrectly explain the result of ViT-S/32-SAM: **Table 2 in [2] (70.5%) reports the results of Inception-style preprocessing (with resolution 224) rather than a combination of strong data augmentations** (See the caption of Table 2 in [2]). Also this 70.5% matches to the results of VIT-S/32-SAM in [37] (See Table 1 in [37]).Therefore, I think the authors misunderstand their main references [2, 8, 21, 37] and refer results of these references incorrectly. For the similar reason, credibility of the first row in second Table in A2 should be replenished.

---

> > > > > ### Author Response · Authors · 2022-08-07
> > > > > **Response to additional questions (3/4)**
> > > > >
> > > > > **Q11** On reproducing SAM.
> > > > >
> > > > > **A11** First, we would like to re-emphasize (as we have arleady stated in Line 230 of our submission) that we religiously followed the experimental settings of SAM’s follow-up works [2,21,6,37]. The comparison are thus fair and reliable. Our proposed SAF and MESA have been verified to **improve the efficiency** of the SAM-like methods with comparable or even better generalization performance. Improving the efficiency of SAM to the base optimizer without sacrificing  generalization performance is the main contribution of our submission.
> > > > >
> > > > > In more detail, the main point of our response is also to clarify the fairness of our experimental setting, which is
> > > > >
> > > > > >As we stated on Line 230, we followed the experimental settings of SAM’s follow-up works [2,21,6,37] for fair comparison with them.
> > > > >
> > > > > The difference of the reported accuracy of ResNet 50 on the ImageNet trained by SAM between SAM paper [8] and SAM's follow-up works [2,6,37] are listed as
> > > > >
> > > > > |ResNet 50-SAM in [8]| ResNet 50-SAM in [2] | ResNet 50-SAM  in [6] |ResNet 50-SAM  in [37] |ResNet 50-SAM  in our submission|
> > > > > | :-----------: | :-----------: |:-----------:|:-----------:|:-----------:|
> > > > > |77.5|76.7|76.7|76.9|76.9|
> > > > >
> > > > > Our reported result matches the reported results in [2,6,37] of ResNet 50-SAM up to 0.2. This unavoidable slight discrepancy is due to randomness of initialization and the selection of batches at each gradient descent step.
> > > > >
> > > > > The reason SAM [8] reports a higher accuracy (77.5) has been stated in our response.
> > > > >
> > > > > >More specifically, SAM uses 100 epochs (vs 90 epochs in our setting) and SAM uses label smoothing of 0.1 (vs no label smoothing in our setting). Thus SAM reports 77.1% accuracy with  ResNet 50 SGD while [2,21,6,37] and our paper report 76.0% accuracy. The mentioned paper  [2]  (cited as [2] in our submission) is the first work to use such different settings than SAM.
> > > > >
> > > > > As we also claimed in Appendix A.4, we strictly follow the setting of [2,21,6,37] to use basic data augmentaion to resize and crop images to 224-pixel resolution, normalize them,  cosine learning rate schedule, SGD optimizer with momentum 0.9, weight decay 0.0001, **no label smoothing** and **only 90 epochs for ResNets**, 300 epochs for ViT.
> > > > >
> > > > >
> > > > > Second, our result of ViT-S/32-SAM matches the reported results in the LookSAM paper [21] (exactly 68.9 vs 68.9!). Under the same setting, our proposed SAF and MESA have been **verified to be more efficient** (Training speed SAF 5,108 img/s vs 4,273 img/s LookSAM) and **achieve better performance** (SAF 69.5 vs 68.8 LookSAM) than LookSAM.
> > > > >
> > > > > To be continued in response to the additional questions (4/4)

---

> > > > > > ### Author Response · Authors · 2022-08-07
> > > > > > **Response to additional questions (4/4)**
> > > > > >
> > > > > > Continued of A11
> > > > > >
> > > > > > For the gap between the reported results of [2,37] and [21], the reported results of ViT-S/32-SAM are precisely listed as
> > > > > >
> > > > > > |Vit-S/32-SAM Reported in [2] | Vit-S/32-SAM Reported in [37] |Vit-S/32-SAM Reported in [21] |Vit-S/32-SAM Reported  in our submission|
> > > > > > | :-----------: | :-----------: |:-----------:|:-----------:|
> > > > > > |70.5|70.5|68.9|68.9|
> > > > > >
> > > > > > As we stated in our response,
> > > > > >
> > > > > > >We carefully reproduced the ViT-SAM results using only basic data augmentation (inception style) and only obtained 68.9% accuracy for ViT S/32, which **exactly** matches the reported results in LookSAM paper [21].
> > > > > >
> > > > > > The most plausible reason why the 70.5 in [2,37] are higher than 68.9 in [21] and our submission is the jax framework and TPU difference. We consulted the authors of [37] via email during the rebuttal period to ask about the implementation of ViT-S/32. The author of [37] replied us as follow,
> > > > > >
> > > > > > >Another tricky part is my experiment used AdamW whose weight decay is multiplied by learning rate, but seems the latest jax implementation fixs weight decay (not multiply by lr), so you would need to take care with this.
> > > > > >  Another tricky part is, SAM/GSAM is per-worker (per-GPU) perturbation, which means the perturbation based on gradient is different across GPUs. So in PyTorch it means the gradient is not sychronized until you finished the second pass of backward; in jax grad is not synchronized by default unless you call "jax.pmean"; but in PyTorch, I think it's synchronized by default unless you do some specific modifications.
> > > > > >  Because grad is not synchronized, the number of workers and n_samples_per_worker is important, so you might need to check both batchsize and n_TPU_cores (8x8 in my eperiment) in the appendix. For example, with 8x8 TPUs, fix batchsize=4096, each worker has 4096//64=64 images; with 4x4 TPUs, this number is 256, and the result could be worse.
> > > > > >
> > > > > > | |Framework (AdamW trick)|TPU or GPU (m-sharpness trick)|
> > > > > > | ----------- | :-----------: |:-----------:|
> > > > > > |[37]|Jax| 8x8 TPUs |
> > > > > > |Ours|Pytorch|8 GPUs |
> > > > > >
> > > > > > The two tricks explain the difference of Vit-S/32-SAM's performance between [2,37] with [21] and ours.
> > > > > >
> > > > > > Last, we have typos in the first table of our response A1. We sincerely and unreservedly apologize for this. The ResNet-50-SAM result should be 76.9 instead of 76.0. In our submission, we report the 76.9 for ResNet-50-SAM in Table 1 of Page 7. Furthermore, the entries 77.5 (SAM reported) and 70.5 [2] should not under the column of strong data augmentation. However, the results of the table in A2 are under the same setting of [2,6,37]; this ensure fair comparison throughout.

---

> > > > ### Comment · Reviewer_RMT8 · 2022-08-06
> > > > **Additional questions - 2**
> > > >
> > > > * **On the derivation of L164-L165**
> > > >
> > > >   While authors argued that $\arg\min_{\theta_t}R_{\mathbb{B_t}}(f_{\theta_t})=\arg\min_{\theta_t}\gamma_{t} R_{\mathbb{B_t}}(f_{\theta_t}) R_{\mathbb{B_t}}(f_{\theta_t})=\arg\min_{\theta_t}[\gamma_t R_{\mathbb{B_t}}(f_{\theta_t}) R_{\mathbb{B_t}}(f_{\theta_t})+\gamma_i R_{\mathbb{B_t}}(f_{\theta_i}) R_{\mathbb{B_i}}(f_{\theta_i})]$ in A3, There exist several issues in this equations.
> > > >
> > > >   * This equation is not equivalent to the original equation in L164-165 $\arg\min_{\theta_t}R_{\mathbb{B_t}}(f_{\theta_t})=\arg\min_{\theta_t}\underset{\theta_i \sim \operatorname{Unif}(\{\theta_{1}, \theta_{2}, \ldots, \theta_{t}\})}{\mathbb{E}}[\gamma_i R_{\mathbb{B_t}}(f_{\theta_i}) R_{\mathbb{B_i}}(f_{\theta_i})]$. Where is expectation term ($\underset{\theta_i \sim \operatorname{Unif}(\{\theta_{1}, \theta_{2}, \ldots, \theta_{t}\})}{\mathbb{E}}$) is your answer? Also, the product $\gamma_t R_{\mathbb{B_t}}(f_{\theta_t}) R_{\mathbb{B_t}}(f_{\theta_t})$ is not equivalent to the product term in the original paper($\gamma_i R_{\mathbb{B_t}}(f_{\theta_i}) R_{\mathbb{B_i}}(f_{\theta_i})$) since $\gamma_i\neq \gamma_t$ in general. Therefore, I think there is still **gap between their answer and the derivation of equation** $\arg\min_{\theta_t}R_{\mathbb{B_t}}(f_{\theta_t})=\arg\min_{\theta_t}\underset{\theta_i \sim \operatorname{Unif}(\{\theta_{1}, \theta_{2}, \ldots, \theta_{t}\})}{\mathbb{E}}[\gamma_i R_{\mathbb{B_t}}(f_{\theta_i}) R_{\mathbb{B_i}}(f_{\theta_i})]$.
> > > >   * Furthermore, $\arg\min_{\theta_t}R_{\mathbb{B_t} }(f_{\theta_t})=\arg\min_{\theta_t}\gamma_{t} R_{\mathbb{B_t}}(f_{\theta_t}) R_{\mathbb{B_t}}(f_{\theta_t})$ in their answer have an error: Since author defined $\gamma_t := \frac{\eta_t}{\rho^2} \cos(\Phi_t)$, $\gamma_t = \frac{\eta_t}{\rho^2}$ as the angle between the gradient using the same mini-batch should 1 (Also, I would recommend authors to define $\cos(\Phi_t)$ in mathematical notation to prevent ambiguity for readers (e.g., State of the parameter($\theta_i? \theta_t?$) is not clear to compute the gradient)). By plugging this to the equation $\arg\min_{\theta_t}R_{\mathbb{B_t} }(f_{\theta_t})=\arg\min_{\theta_t}\gamma_{t} R_{\mathbb{B_t}}(f_{\theta_t}) R_{\mathbb{B_t}}(f_{\theta_t})$, $\arg\min_{\theta_t}R_{\mathbb{B_t} }(f_{\theta_t})=\arg\min_{\theta_t}\gamma_{t} R_{\mathbb{B_t}}(f_{\theta_t}) R_{\mathbb{B_t}}(f_{\theta_t}) = \arg\min_{\theta_t}\frac{\eta_t}{\rho^2}R_{\mathbb{B_t}}(f_{\theta_t})^2$ which means $R_{\mathbb{B_t} }(f_{\theta_t}) = \frac{\eta_t}{\rho^2}R_{\mathbb{B_t}}(f_{\theta_t})^2 arrow R_{\mathbb{B_t} }(f_{\theta_t}) = \frac{\rho^2}{\eta_t}$. However, **the last equation does not hold in general** since $\rho$ is the radius of the neighborhood and $\eta_t$ is the learning rate that is defined as arbitrary.
> > > >   * Also, equation $\arg\min_{\theta_t}\gamma_t R_{\mathbb{B_t}}(f_{\theta_t}) R_{\mathbb{B_t}}(f_{\theta_t})=\arg\min_{\theta_t}[\gamma_t R_{\mathbb{B_t}}(f_{\theta_t}) R_{\mathbb{B_t}}(f_{\theta_t})+\gamma_i R_{\mathbb{B_t}}(f_{\theta_i}) R_{\mathbb{B_i}}(f_{\theta_i})]$ is not clear, neither. Why $\gamma_i R_{\mathbb{B_t}}(f_{\theta_i}) R_{\mathbb{B_i}}(f_{\theta_i})$ is constant? Is there **any derivation of this argument** in the original paper or their answer?
> > > >
> > > >
> > > >
> > > > * **On A4**
> > > >   * The answer does not explain 'why SAF/MESA found flatter local minima **than SAM**: They only explain the motivation (as shown in Fig. 3) behind SAF and MESA. Although this motivation can explain how SAF and MESA can find the flat minima, this motivation does not explain SAF and MESA find flatter minima than SAM. Is there any specific case (even for Toy example) where SAM would fail but SAF/MESA would not? Concrete numerical examples (even in low-dimensional cases) would help the readers' understanding.

---

> > > > > ### Author Response · Authors · 2022-08-07
> > > > > **Response to additional questions (1/4)**
> > > > >
> > > > > **Q6** This equation is not equivalent to the original equation in L164-165 $\underset{\theta_t}{\arg\min}R_{\mathbb{B}\_t}(f_{\theta_t}) = \underset{\theta_i \sim \mathrm{Unif}( \Theta)}
> > > > > {\mathop{\mathbb{E}}}[ \gamma_i R_{\mathbb{B}\_t}(f_{\theta_i})R_{\mathbb{B}\_i}(f_{\theta_i})]$.
> > > > >
> > > > > **A6**  Our response first stated that
> > > > > >$\underset{\theta_t}{\arg\min}R_{\mathbb{B}\_t}(f_{\theta_t})=\underset{\theta_t}{\arg\min}[ \gamma_t R_{\mathbb{B}\_t}(f_{\theta_t})R_{\mathbb{B}\_t}(f_{\theta_t})+ \gamma_i R_{\mathbb{B}\_t}(f_{\theta_i})R_{\mathbb{B}\_i}(f_{\theta_i})]$.
> > > > >
> > > > > This holds as the added term does not involve $\theta_t$ (it only involves $\theta_i$ and $i\ne t$).
> > > > >
> > > > > Then we stated in our response
> > > > > >"Note that for $i \ne t$,  $\gamma_i R_{\mathbb{B}\_t}(f_{\theta_i})R_{\mathbb{B}\_i}(f_{\theta_i})$ is a constant with respect to the variable $\theta_t$ as it does not involve $\theta_i$. Therefore, we enummerate all the terms $\gamma_i R_{\mathbb{B}\_t}(f_{\theta_i})R_{\mathbb{B}\_i}(f_{\theta_i})$ that satisfy $i \neq t$, and thus we obtain the LHS of Equation (7) "
> > > > >
> > > > > Specifically, $\underset{\theta_t}{\arg\min}R_{\mathbb{B}\_t}(f_{\theta_t})=\underset{\theta_t}{\arg\min}[ \gamma_t R_{\mathbb{B}\_t}(f_{\theta_t})R_{\mathbb{B}\_t}(f_{\theta_t})+ \sum_{i<t} \gamma_i R_{\mathbb{B}\_t}(f_{\theta_i})R_{\mathbb{B}\_i}(f_{\theta_i})]$ = $\underset{\theta_t}{\arg\min} \sum_{i=1}^t \gamma_i R_{\mathbb{B}\_t}(f_{\theta_i})R_{\mathbb{B}\_i}(f_{\theta_i})=\underset{\theta_t}{\arg\min} \underset{\theta_i \sim \mathrm{Unif}( \Theta)}
> > > > > {\mathop{\mathbb{E}}}[ \gamma_i R_{\mathbb{B}\_t}(f_{\theta_i})R_{\mathbb{B}\_i}(f_{\theta_i})]$.
> > > > >
> > > > > This gives the LHS of Equation (7)  $\underset{\theta_i \sim \mathrm{Unif}( \Theta)}
> > > > > {\mathop{\mathbb{E}}}[ \gamma_i R_{\mathbb{B}\_t}(f_{\theta_i})R_{\mathbb{B}\_i}(f_{\theta_i})]$. That is how we derive $\underset{\theta_t}{\arg\min}[ \gamma_t R_{\mathbb{B}\_t}(f_{\theta_t})R_{\mathbb{B}\_t}(f_{\theta_t})+\gamma_i R_{\mathbb{B}\_t}(f_{\theta_i})R_{\mathbb{B}\_i}(f_{\theta_i})]=\underset{\theta_t}{\arg\min} \underset{\theta_i \sim \mathrm{Unif}( \Theta)}
> > > > > {\mathop{\mathbb{E}}}[ \gamma_i R_{\mathbb{B}\_t}(f_{\theta_i})R_{\mathbb{B}\_i}(f_{\theta_i})]$
> > > > >
> > > > > Therefore, it is equivalent to the original equation, which is
> > > > > > $\underset{\theta_t}{\arg\min}R_{\mathbb{B}\_t}(f_{\theta_t}) = \underset{\theta_i \sim \mathrm{Unif}( \Theta)}
> > > > > {\mathop{\mathbb{E}}}[ \gamma_i R_{\mathbb{B}\_t}(f_{\theta_i})R_{\mathbb{B}\_i}(f_{\theta_i})]$.
> > > > >
> > > > >
> > > > >
> > > > > ---
> > > > > **Q7** Also, the product $\gamma_t R_{\mathbb{B}\_t}(f_{\theta_t})R_{\mathbb{B}\_t}(f_{\theta_t})$ is not equivalent to the product term in the original paper $\gamma_i R_{\mathbb{B}\_t}(f_{\theta_i})R_{\mathbb{B}\_i}(f_{\theta_i})$ since $\gamma_i \ne \gamma_t$ in general.
> > > > >
> > > > > **A7** Throughout the submission, we use $t$ to index the current model update step, and $i < t$ to index the pervious steps. The natural number **$i$ represents an iteration index**, which indexes $\theta_i \sim \mathrm{Unif}( \Theta)$ in Equation (7) of our submission. When the iteration index $i$ arrives at $i=t$ ($t$ is the current iteration and $i$ is a previous iteration), the term $\gamma_i R_{\mathbb{B}\_t}(f_{\theta_i})R_{\mathbb{B}\_i}(f_{\theta_i})$ is  clearly equivalent to the term $\gamma_t R_{\mathbb{B}\_t}(f_{\theta_t})R_{\mathbb{B}\_t}(f_{\theta_t})$.
> > > > >
> > > > > ---
> > > > > **Q8** Furthermore, $\underset{\theta_t}{\arg\min}R_{\mathbb{B}\_t}(f_{\theta_t})=\underset{\theta_t}{\arg\min}[ \gamma_t R_{\mathbb{B}\_t}(f_{\theta_t})R_{\mathbb{B}\_t}(f_{\theta_t})]$ in their answer have an error.
> > > > >
> > > > > **A8** The esteemed reviewer stated that
> > > > > > $\underset{\theta_t}{\arg\min}R_{\mathbb{B}\_t}(f_{\theta_t})=\underset{\theta_t}{\arg\min} \gamma_t R_{\mathbb{B}\_t}(f_{\theta_t})R_{\mathbb{B}\_t}(f_{\theta_t})=\underset{\theta_t}{\arg\min} \frac{\eta_t}{\rho^2} R_{\mathbb{B}\_t}(f_{\theta_t})^2$ which means $R_{\mathbb{B}\_t}(f_{\theta_t})=\frac{\eta_t}{\rho^2} R_{\mathbb{B}\_t}(f_{\theta_t})^2$
> > > > >
> > > > > However, the equation you claimed "$R_{\mathbb{B}\_t}(f_{\theta_t})=\frac{\eta_t}{\rho^2} R_{\mathbb{B}\_t}(f_{\theta_t})^2$" **does not hold**. Note that the equation we claimed "$\underset{\theta_t}{\arg\min}R_{\mathbb{B}\_t}(f_{\theta_t})=...=\underset{\theta_t}{\arg\min} \frac{\eta_t}{\rho^2} R_{\mathbb{B}\_t}(f_{\theta_t})^2$" is to find the solution of **arg min**. The equation does not hold without the "**arg min**". One cannot "cancel" $\arg\min$ on both sides of any equation. Therefore, the equation you derived is not correct, and we do not have error in this part, to the best of our knowledge.

---

> > > > > > ### Author Response · Authors · 2022-08-07
> > > > > > **Response to additional questions (2/4)**
> > > > > >
> > > > > > **Q9** Why $\gamma_i R_{\mathbb{B}\_t}(f_{\theta_i})R_{\mathbb{B}\_i}(f_{\theta_i})$ is a constant?  Is there any derivation of this argument in the original paper or their answer?
> > > > > >
> > > > > > **A9** We stated in our reponse "$\gamma_i R_{\mathbb{B}\_t}(f_{\theta_i})R_{\mathbb{B}\_i}
> > > > > > (f_{\theta_i})$ is a constant **with respect to the variable $\theta_t$**". This is clearly true as $\gamma_i R_{\mathbb{B}\_t}(f_{\theta_i})R_{\mathbb{B}\_i}
> > > > > > (f_{\theta_i})$ does not involve the term $\theta_t$.
> > > > > >
> > > > > > We **did not claim** "$\gamma_i R_{\mathbb{B}\_t}(f_{\theta_i})R_{\mathbb{B}\_i}(f_{\theta_i})$ is a constant". It clearly depends on some variables such as $\gamma_i$ and $\theta_i$.
> > > > > >
> > > > > > ---
> > > > > >
> > > > > > **Q10** On A4 The answer does not explain 'why SAF/MESA found flatter local minima than SAM
> > > > > >
> > > > > > **A10** We **did not claim** either in our submission or our response that "SAF/MESA found flatter local minima than SAM". What we claimed in our submision in lines 281 is
> > > > > >
> > > > > > >The visualized loss landscape clearly demonstrate that SAF can converge to a region as flat as SAM does.
> > > > > >
> > > > > > What we claimed in our response A5 is
> > > > > >
> > > > > > >and showed that SAF’s loss landscape is as flat as MESA’s.
> > > > > >
> > > > > > Your question and claim that "why SAF/MESA found flatter local minima than SAM" is thus not truly reflective of our claim. We didn't make this claim. Pictorially, we show that SAF/MESA's loss landscapes are as flat at SAM's. We have updated Figure 1 in Appendix A3 to show that SAF/MESA find local minima which are **as flat as SAM's**.
> > > > > >
> > > > > >
> > > > > >
> > > > > > We re-emphasize here that the motivation and main contribution of this submission is to improve the efficiency issue of SAM. We do not claim to find flatter local minima than SAM or its variants.
> > > > > >
> > > > > > ---

---

> > ### Public Comment · ~Tao_Li12 · 2023-01-27
> > **Questions about the hyper-parameters of SAF with stronger data augmentation**
> >
> > Hi Authors,
> >
> > Currently, I am conducting some related reproducing experiments. Could you kindly provide the detailed setting of SAF with stronger data augmentation (e.g. training epochs, hyperparameters for mixup and cutmix)?  Many thanks and looking to your response:)
> >
> > Sincerely,
> > Tao

---

> > > ### Public Comment · ~Jiawei_Du1 · 2023-01-27
> > > **Parameters setting**
> > >
> > > Hi tao,
> > >      we have uploaded the codes for reproduction in https://github.com/AngusDujw/SAF. You may refer to it for reproduction. We also append the parameters setting in the appendix of our paper (pages 16-17).
> > >      We take the imagenet dataset as an example. We use training epoch 90 for resnets and 300 for the vit. We do not use any strong data augmentation techs such as mixup or cutmix, we only use inceptional style data augmention only. You may email me if you have more questions in reproduction.
> > > Best regards
> > > Jiawei

---

> > > > ### Public Comment · ~Tao_Li12 · 2023-01-27
> > > > **Further questions**
> > > >
> > > > Hi Jiawei,
> > > >
> > > > Thanks for your instant reply. I mean the experiments in the response, e.g. **A2**, as the training details are not reported. Besides, I have sent an email to you for your checking.
> > > >
> > > > Best wishes,
> > > > Tao

---

> > > > ### Public Comment · ~Tao_Li12 · 2023-02-01
> > > > **The mail has been sent.**
> > > >
> > > > Hi Jiawei,
> > > >
> > > > I have sent the mail to dujiawei@u.nus.edu and also received your mail. I think my mail may be filtered by the mail system (may check the SPAM).
> > > >
> > > > Best wishes,
> > > > Tao

---

### Official Review · Reviewer_Byfm · 2022-07-10

**Rating:** 7
**Confidence:** 4
**Soundness:** 3 good
**Presentation:** 4 excellent
**Contribution:** 4 excellent

**Summary:**

This paper proposes a sharpness-aware minimization method that adds almost no computational overhead. The authors propose to use the training loss differences (more precisely, the KL divergence of output predictions) to proxy the sharpness term. The advantage of using this new metric would be that it only requires additional memory but no additional computations. They also propose an adjusted method that addresses the memory problem of the first method. The effectiveness of the proposed methods is then shown experimentally for Imagenet and CIFAR datasets.

**Questions:**

1- It is mentioned that before epoch E_start the outputs are not stable. What would happen if from the first epoch this method is used? Does this instability affect the performance?

2- How should the hyper-parameters (tau, E, and lambda) be selected for your method? Have you studied various hyper-parameters? It is mentioned that a fixed value is used for these hyper-parameters.

3- In Figure 4b, why does MESA have higher sharpness at the beginning of training but lower sharpness at the end of training compared to SAF? Also, from the visualization, it appears that MESA has the flattest landscape.


**Limitations:**

The limitation of the method with regards to memory is mentioned and properly addressed.


**Strengths And Weaknesses:**

Strengths:

1- The paper is very well-written.

2- The limitation of the initial method in terms of memory has been properly addressed in the follow-up method.

3- The method is simple and efficient to use.

4- The experiments use state-of-the-art settings.

Weaknesses:

1- The method doesn't always outperform SAM in terms of accuracy. This is not so much a weakness because the method is mainly intended to improve computational overhead.

2- The experiments are only for image-classification tasks.

3- The method introduces a few hyper-parameters.

---

> ### Author Response · Authors · 2022-08-02
> **Response**
>
> Thank you for your constructive comments! We give point-to-point replies to your questions in the following.
>
>
>
> ---
>
> **Q1:**  It is mentioned that before epoch E_start the outputs are not stable. What would happen if from the first epoch this method is used? Does this instability affect the performance?
>
> **A1:** In the submission, we used $E_\mathrm{start}  = 5$ for both SAF and MESA on the CIFAR datasets. It is worth noting that computing the proposed trajectory loss requires information from previous training epochs. Hence our method is **not** applicable for the first few epochs.  Following your kind suggestion, we applied our method from the earliest valid training epochs. Specifically, the earliest epoch of  SAF is $E_\mathrm{start}  = 4$, as SAF takes the output in $\tilde{E}=3$ epochs ago to compute the trajectory loss. The earliest epoch of MESA is $E_\mathrm{start}  = 1$, as MESA takes the output of the Exponential Moving Average (EMA) model to compute the trajectory loss.
>
> With the above setting, we conducted additional experiments with ResNet18 to investigate model's performance with different $E_\mathrm{start}$'s. The results are stated below.
>
> | |CIFAR-10|CIFAR-100 |
> | ----------- | :-----------: |:-----------:|
> |SAF ($E_\mathrm{start}$ = 4)  | 96.35 $\pm$ 0.04|80.11 $\pm$ 0.07|
> |SAF ($E_\mathrm{start}$ = 5)  | 96.37 $\pm$ 0.02|80.06 $\pm$ 0.05|
> |MESA ($E_\mathrm{start}$ = 1) | 96.18 $\pm$ 0.06|79.54 $\pm$ 0.04|
> |MESA ($E_\mathrm{start}$ = 5) | 96.24 $\pm$ 0.02|79.79 $\pm$ 0.09|
>
> The above experimental results demonstrate that the instability affects the final performance only  marginally. MESA’s performance decreased by 0.25% on the CIFAR 100 dataset but is unchanged on the CIFAR 10 dataset.  We will add these experimental results in our updated version.
>
>
>
> ---
>
> **Q2:** How should the hyper-parameters ($\tau$, E, and $\lambda$) be selected for your method?  The method introduces a few hyper-parameters.
>
> **A2:** We choose these optimal parameters via standard grid search on an isolated CIFAR-100 validation set. This has been detailed in Appendix A.3. We find some parameters ($\tau$, $\tilde{E}$, $E_\mathrm{start}$) are consistent among different architectures and datasets, and only the coefficient $\lambda$ needs to be tuned for different architectures and datasets.
>
>
> ---
>
> **Q3:**  In Figure 4b, why does MESA have higher sharpness at the beginning of training but lower sharpness at the end of training compared to SAF? Also, from the visualization, it appears that MESA has the flattest landscape.
>
> **A3:**  As explained in Lines 203 and 204,  MESA computs the trajectory loss using the output of the EMA model, which averages the past model weights in an exponentially decaying fashion and thus weighs the more recent weights higher. Thus, compared with SAF, the sharpness of MESA will be affected more by the latest model that is updated by  vanilla SGD. Therefore, when the sharpness of the vanilla (SGD) model is increasing/decreasing (resp.), MESA’s sharpness would be higher/lower (resp.) than that of SAF’s.
>
> As for the visualization that MESA appears to have the flattest landscape, this is because to produce the  visualization, we (and others) use a large amount of Gaussian perturbation in the parameter space. Thus, the visulization involves not only the central landscape around the converged minimum, but also the landscape in **several epochs** ago. As we can see in Figure 4b of our submission, MESA's sharpness is lower than SAF's from epochs 140 to 185. However, only the sharpness of the central landscape **around the converged minimum** is more predicative of the generalization ability. We also visualized the loss landscape with a tiny amount (value of 0.07 vs 0.2) of Gaussian perturbation to visualize the central landscape around the converged minimum, and showed that SAF’s loss landscape is as flat as MESA’s. We have updated the visualization with a tiny amount of adversarial perturbation in Figure 1 of the appendix.

---

> > ### Comment · Reviewer_Byfm · 2022-08-08
> > **Thanks for the response**
> >
> > Thanks for the response and the additional experiments.

---

### Meta-Review · Area_Chair_AjpH · 2022-08-30

**Recommendation:** Accept
**Confidence:** Less certain

**Metareview:**

This paper proposes a novel optimization method, called SAF, for reaching flat minima. The main claim is that the proposed method does not suffer from computational overhead of SAM-like methods which is typically 2x SGD. The proposed method is based on a novel loss that minimizes the KL-divergence between the output of the network with previous weights and current weights. This allows avoiding the extra computational overhead of SAM. Authors show that their method can achieve better empirical results in a compute-constrained regime.

Reviewers are in agreement about the novelty of the proposed method and that the paper is well-written and easy to follow. The empirical results also show a clear advantage over other methods in a compute-constrained regime. The main concern about accepting the paper is due to some mismatch between reported numbers in the paper and that of original SAM paper as well as lack of clarity on some experimental details. I am leaning towards acceptance given the advantages mentioned above but I strongly recommend authors (and want to see this implemented for camera-ready) to at the very least make the following changes (as well as the ones proposed by reviewers) to adhere to clarity and reproducibility standards of publications in computer science and increase the impact of their paper:

1- In Table 1 (and perhaps Figure 1?), add ALL reported results for SAM including but not limited to a) The results reported in the original SAM paper. b) The results reported by other papers running SAM themselves c) The results of running SAM by authors.

2- In Table 1 (and perhaps everywhere), always make it extra easy for the reader to know if the number is taken from another paper or it is a the result you reproduced yourself.

3- Add all experimental details (including augmentation techniques used and what hyper-parameters are tuned) for all experiments. When there are mismatches that makes the comparison more difficult, this becomes even more important and allows the reader to make-up their mind about where the improvement might be coming from.

4- If you have considerations about original SAM paper’s numbers not being reproduced by other papers, you can add it as a footnote or discussion but it is still important to report them.

**Award:**

No

---

### Decision · Program_Chairs · 2022-09-14

Accept